# Multifaceted effects of variable biotic interactions on population stability in complex interaction webs
Koya Hashimoto [1,2,3,8] ✉, Daisuke Hayasaka [1,8], Yuji Eguchi[4], Yugo Seko[2,4], Ji Cai[5], Kenta Suzuki[6,7], Koichi Goka [2] & Taku Kadoya[2,8]

Recent studies have revealed that biotic interactions in ecological communities vary over time, possibly mediating community responses to anthropogenic disturbances. This study investigated the heterogeneity of such variability within a real community and its impact on population stability in the face of pesticide application, particularly focusing on density-dependence of the interaction effect. Using outdoor mesocosms with a freshwater community, we found considerable heterogeneity in density-dependent interaction variability among links in the same community. This variability mediated the stability of recipient populations, with negative density-dependent interaction variability stabilizing whereas positive density-dependence and density-independent interaction variability destabilizing populations. Unexpectedly, the mean interaction strength, which is typically considered crucial for stability, had no significant effect, suggesting that how organisms interact on average is insufficient to predict the ecological impacts of pesticides. Our findings emphasize the multifaceted role of interaction variability in predicting the ecological consequences of anthropogenic disturbances such as pesticide application.

Predicting the magnitude to which anthropogenic disturbances affect ecosystems is essential for successful conservation and ecosystem management. In real ecosystems, all members of a biological community are linked to each other by biotic interactions such as predation, competition, and mutualism, and these interactions often cause unpredictable responses of biological communities to natural and/or anthropogenic disturbances[1–3]. Indeed, ecologists have considered that community responses to disturbances highly depend on the strength of biotic interactions: weak interactions are likely to promote various types of community stability against system alterations[4–7]. Notably, many earlier studies have postulated a 'static view' of interaction networks, focusing on long-term average properties of interactions responsible for community stability[8–10]. More recently, however, researchers have recognized that the effect of interactions can be highly variable due to behavioural or physiological trait plasticity and rapid evolution in response to intrinsic and extrinsic factors[11–17]. This view of variable interactions urges us to reconsider how biotic interactions determine community responses to anthropogenic disturbances.

One of the important sources of interaction variability is density-dependence in the *per capita* interaction effect. The *per capita* interaction effect is measured by changes in the *per capita* population growth of an interaction recipient species caused by slight changes in donor species density[18,19]. The density-dependence of *per capita* interaction effects generates nonlinear functional responses that are well known to play a critical role in determining the stability of resource–consumer populations[20,21], in which the variability in the *per capita* interaction effect is an implicit but essential underlying mechanism. Specifically, with negative interaction density-dependence, a decrease in the density of the recipient causes an interaction change that positively affects the recipient such that the collapse of recipient populations is avoided (i.e., stabilizing). Conversely, with positive interaction density-dependence, a decline in a recipient's population provides a change in interaction effect disadvantageous to the recipient population, leading to a greater probability of extinction (i.e., destabilizing). Although the effects of functional response on population stability have been studied mostly in terms of resource–consumer relationships, the

[1]Faculty of Agriculture, Kindai University, Nakamachi 3327-204, Nara, Nara, 631-8505, Japan. [2]National Institute for Environmental Studies (NIES), Onogawa 16-2, Tsukuba, Ibaraki, 305-8506, Japan. [3]Faculty of Agriculture and Life Science, Hirosaki University, 3 Bunkyo-cho, Hirosaki, Aomori, 036-8561, Japan. [4]Graduate School of Agriculture, Kindai University, Nakamachi 3327-204, Nara, Nara, 631-8505, Japan. [5]Center for Ecological Research, Kyoto University, Hirano 2-509-3, Otsu, Shiga, 520-2113, Japan. [6]BioResource Research Center, RIKEN, Takanodai 3-1-1, Tsukuba, Ibaraki, 305-0074, Japan. [7]Institute for Multidisciplinary Sciences, Yokohama National University, Tokiwadai 9-5, Hodogaya, Yokohama, Kanagawa, 240-8501, Japan. [8]These authors contributed equally: Koya Hashimoto, Daisuke Hayasaka, Taku Kadoya. ✉e-mail: atrophaneura4@gmail.com

underlying mechanisms explicitly considering the variability in the interactions described above can be applied to other types of interactions, such as competition and mutualism[22,23].

An important challenge here is understanding the effects of interaction density-dependence in the context of large communities involving numerous species and a complex network of interactions among them. Recent theoretical studies have suggested that it is possible to scale up the stabilizing/destabilizing effects of interaction density-dependence from small numbers of species populations to communities consisting of many species[23,24]. At the same time, they have showed that incorporating heterogeneity of interaction density-dependence among links dramatically alters community stability and species diversity–stability relationships[23]. In particular, the proportion and/or the effect size of stabilizing and destabilizing density-dependence are critical for determining the stability of the whole community because the stabilizing effects of species interactions may cancel out destabilizing interactions when they have a greater proportion and/or larger effects. Thus, quantifying heterogeneity in the forms of density-dependence among links is essential for understanding the stabilizing mechanisms of natural communities. However, describing the heterogeneity of the functional forms in natural communities is logistically challenging: this is because interaction density-dependence results from various mechanisms, such as constraints by prey handling time, optimal foraging, and adaptive defence and because these underlying mechanisms are shaped by the unique ecological and evolutionary context that each interaction link has experienced.

Here, we explored (i) heterogeneity in interaction density-dependence of interaction links within a real community and (ii) whether interaction density-dependence works to stabilize/destabilize populations in response to anthropogenic disturbances. We combined a manipulative open mesocosm experiment and nonlinear time-series analysis to track variability in interaction effects. As a model system, we chose pesticide and freshwater communities in paddy ponds, which are representative agricultural habitats in East Asia. We conducted three-year outdoor mesocosm experiments in which the ponds were fully crossed with two replicates each of an insecticide (fipronil) and an herbicide (pentoxazone). During the experimental period (approximately 140 days) of each year, we monitored the densities of ten community members, including phyto- and zooplankton, plants, and macroinvertebrates (Supplementary Table 1), every two weeks. Based on the fortnight-interval data, we quantified interaction effects among these community members and their variability by empirical dynamic modelling

(EDM), which is a recently developed analytical framework for nonlinear time-series data[25,26].

Using the EDM approach, we reconstructed the interaction network and its density-dependence of the communities in the paddies and distinguished observed interaction variability into three types in terms of recipient density-dependence in the *per capita* interaction effect (hereafter referred to as interaction density-dependence; IDD): (1) negative IDD, (2) positive IDD, and (3) density-independent interaction variability (see Methods for details). We then tested the following three specific predictions: (1) a negative IDD increases the stability of the recipient population because a decrease or increase in the recipient density is compensated for by changes in the interaction effect that are positive or negative to the recipient, respectively (Fig. 1a); (2) a positive IDD increases the instability of the recipient population because a decrease or increase in the recipient density is magnified by changes in the interaction effect that are negative or positive to the recipient, respectively (Fig. 1b); and (3) density-independent interaction variability is more likely to destabilize the recipient population because larger density-independent interaction changes induce greater unpredictable variability in the recipient population (Fig. 1c).

## Results

### Pesticide impacts on the density of each community member

Here, we describe whether and how the three pesticide treatments (i.e., I: insecticide alone, H: herbicide alone, and I + H: insecticide and herbicide) affected the density of the ten community members in the experimental paddies. For the majority of the community members, insecticide applications had stronger impacts than herbicide applications on their density (Fig. 2a, Supplementary Table 2). Insecticide treatment dramatically decreased the density of phytophilous and benthic predatory insects (i.e., Odonata larvae) [C (control) vs. I (insecticide alone)], which was consistent with previous studies[27,28]. Additionally, although not statistically significant, the insecticide treatment decreased the density of detritivorous insects and increased the density of neustonic predatory insects and molluscs. The herbicide treatment significantly decreased only the density of macrophytes [C vs. H (herbicide alone)]. When we applied both the insecticide and the herbicide, phytoplankton, macrophytes, herbivores, phytophilous predators, and benthic predators significantly decreased in density, and molluscs significantly increased in density [C vs. I + H (mixture of insecticide and herbicide)]. The observed pesticide impacts may have been due to direct toxicity, indirect toxicity via biotic interactions, or a mixture of these factors.

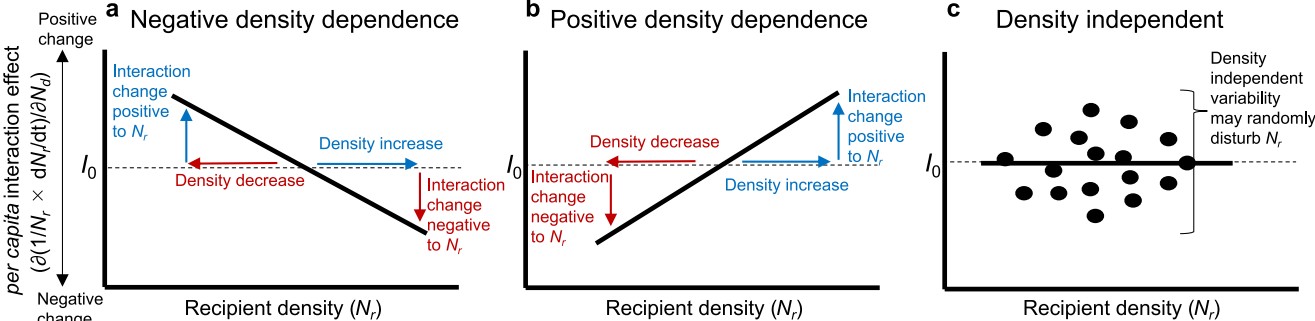

**Fig. 1 | Schematic representation of three types of density-dependent (or density-independent) variability in the per capita interaction effect.** $N_r$ and $N_d$ denote the recipient and donor densities, respectively. The *per capita* interaction effect $(\partial(1/N_r \times dN_r/dt)/\partial N_d)$ may vary negatively or positively depending on recipient density or independent of recipient density. Red and blue arrows indicate density or interaction changes negative or positive to the recipient population, respectively. **a** Negative density-dependence. A decrease or increase in recipient density results in changes in the interaction effect that are positive or negative to the recipient, respectively, resulting in negative feedback and a greater likelihood of stabilizing the recipient population. **b** Positive density-dependence. A decrease or increase in density makes the interaction effect more negative or positive to recipient density,

respectively, bringing positive feedback and potentially leading to population collapse or an outbreak. **c** Density-independent variability. Unlike density-dependent variability in interactions, density-independent interaction changes can cause positive and negative effects on recipient density in an unpredictable manner, resulting in a random disturbance to the recipient population. Note that the solid line in each panel intersects the $I_0$ horizontal lines (*per capita* interaction effect = 0), i.e., the interaction effect changes from positive (negative) to negative (positive), but this is not necessary for interaction variability to have the potential to stabilizing or destabilizing effects. For example, stable coexistence between competitors may arise when competition, which is always negative, becomes stronger (in the downwards direction in the above figures) with increasing density.

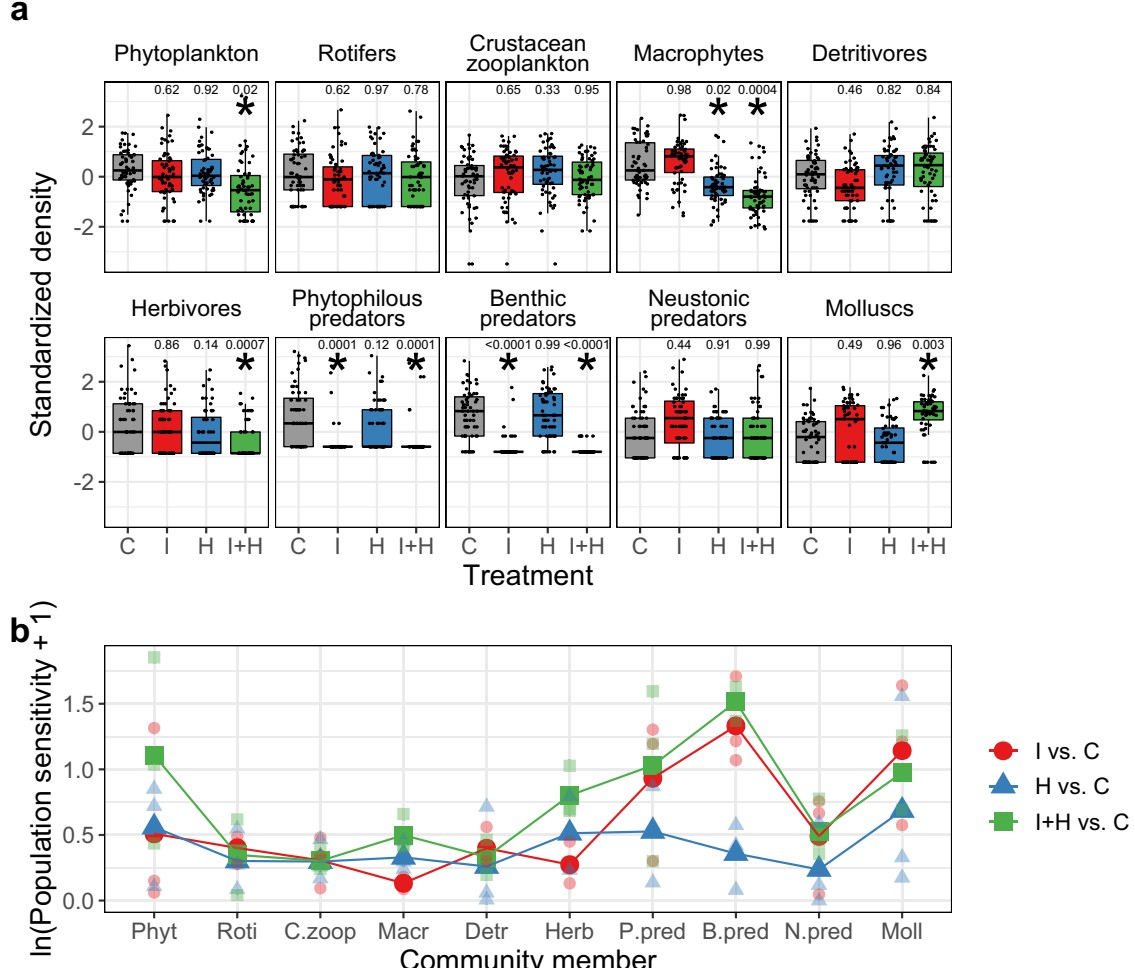

**Fig. 2 | Pesticide impacts on the density of paddy community members.**
**a** Treatment effects on the density of each paddy community member. The treatment abbreviations are as follows: C control, I insecticide alone, H herbicide alone, and I + H: mixture of insecticide and herbicide. Asterisks indicate significant differences compared to the control (Dunnett test, α = 0.05) and exact *P* values of the Dunnett tests are shown above the boxes. Each boxplot is drawn by using the raw values from 6 short time series (2 tank replicates × 3 years) each of which consists of 10 consecutive time points (i.e., *n* = 60). The midline, box limits, and whiskers indicate the median, upper and lower quartiles and 1.5× interquartile range,

respectively. Points indicate the raw values of time series at each time point.
**b** Population sensitivity, which is measured as the absolute value of the log response ratio of the mean raw density (not standardized density) in the pesticide treatments (either I, H or I + H) relative to that in the controls for each of the three experimental years. Large points indicate mean values, while small points indicate raw values. The abbreviations of the community members are as follows: Phyt phytoplankton, Roti rotifers, C.zoop crustacean zooplankton, Macr macrophytes, Detr detritivores, Herb herbivores, P.pred phytophilous predators, B.pred benthic predators, N.pred neustonic predators, and Moll molluscs.

We evaluated population sensitivity to pesticide treatments, which is a proxy of population instability, by calculating the absolute value of the log response ratio (LRR) of the mean density in a pesticide treatment relative to that in the controls[29]. We calculated $LRR_{i,j}$ for community member *i* in year *j* for every pesticide treatment as $\ln((T_{i,j} + 0.1)/(C_{i,j} + 0.1))$, where $T_{i,j}$ is the mean raw density (not standardized density) in either the I, H, or I + H treatment and $C_{i,j}$ is the mean raw density of the controls. We added 0.1 to the numerator and denominator because there were several zero data points for both $T_{i,j}$ and $C_{i,j}$[30]. We found that there was considerable variation in sensitivity among community members (Fig. 2b). That is, some members were sensitive (unstable), and other members were relatively less sensitive (stable) to pesticide disturbance. In particular, rotifers, crustacean zooplankton, and detritivores were relatively less sensitive to all three pesticide treatments. Additionally, this variation in the sensitivity among different community members changed depending on the three pesticide treatments. Phytophilous, benthic, and neustonic predators and molluscs were more sensitive to the I and I + H treatments than to the H treatment, while macrophytes and herbivores were more sensitive to the H and I + H treatments than to the I treatment.

**Heterogeneity in interaction variability among different links**

Our EDM analyses successfully reconstructed the interaction network among paddy community members in each treatment (Fig. 3, Supplementary Table 3, Supplementary Fig. 1). Figure 3a shows the interaction networks in the controls. Note that the arrows represent the net effects of interactions (i.e., the sum of direct and indirect effects) rather than the sole direct effects. Most of the interactions were biologically interpretable; for instance, phytoplankton positively affected rotifers, and rotifers negatively affected phytoplankton but positively affected crustacean zooplankton, suggesting prey–predator interactions (Fig. 3a, Supplementary Note 1). As several previous studies have indicated[5], the distribution of the mean interaction strength given no disturbances was skewed towards a weaker strength (Fig. 3b), and this tendency did not change for the pesticide application treatments (Supplementary Fig. 2). For every treatment, the numbers of negative and positive interactions were similar (Supplementary Fig. 2). Furthermore, the interaction effect varied over time within each year even without pesticides (Fig. 3c, Supplementary Fig. 3). Notably, the magnitude of interaction temporal variability differed greatly among interaction pairs (Fig. 3c). For example, the interaction effect on herbivorous insects

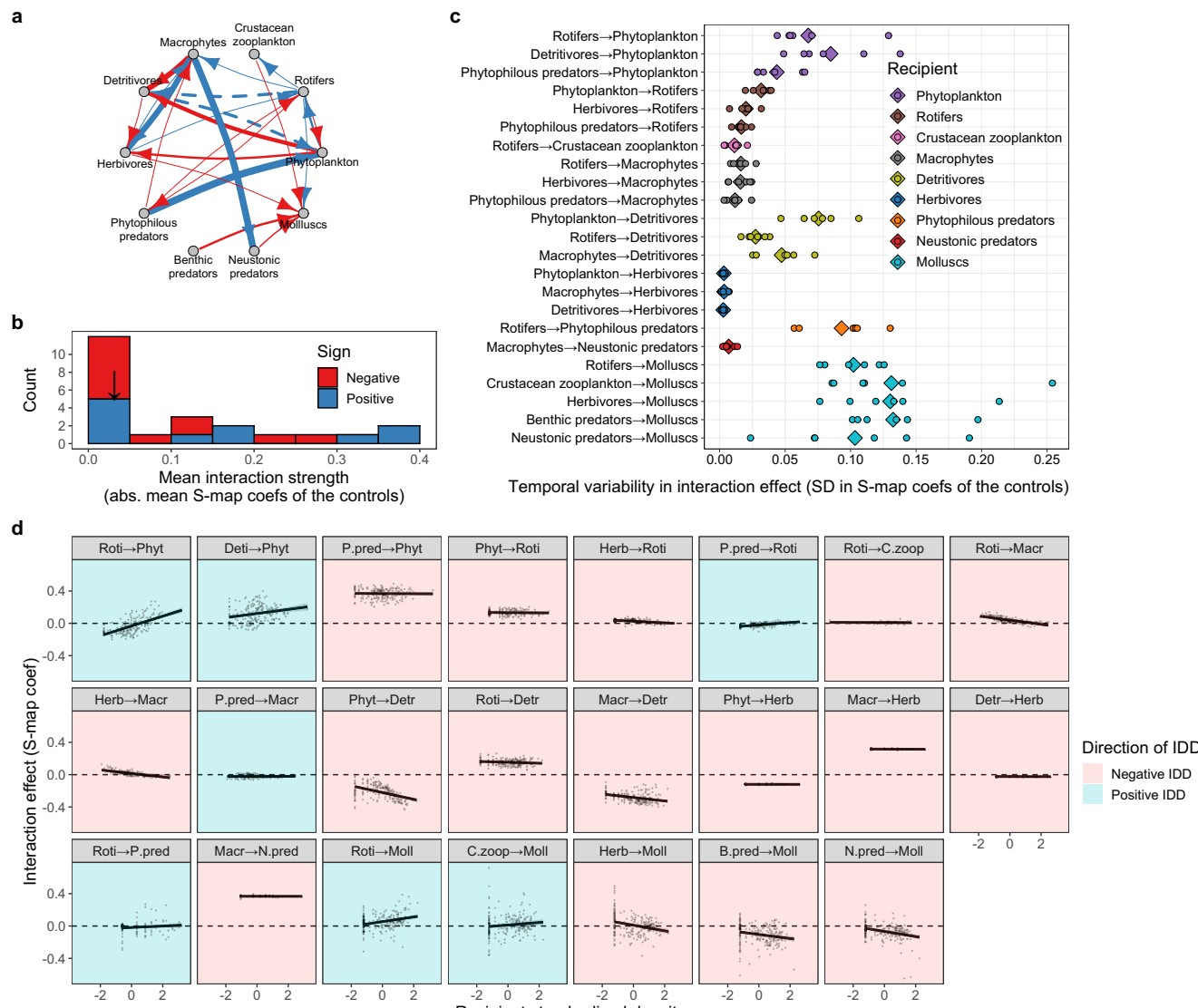

**Fig. 3 | Heterogeneous variability in *per capita* interaction effects within the experimental paddy community. a** Interaction networks of the controls (i.e., treatments without pesticide disturbances) reconstructed by EDM analysis. Red and blue arrows indicate negative and positive interactions, respectively. Their thickness is proportional to the mean interaction strength represented by the absolute values of the mean S-map coefficient averaged over all the experimental periods and replicates. Solid arrows: $P < 0.05$, dashed arrows: $0.05 < P < 0.1$. **b** Right-skewed distribution of the mean *per capita* interaction strength in the control treatments (absolute values of mean S-map coefficients averaged over all the experimental periods and replicates in the controls) represented by a histogram. The vertical arrow indicates the median values. **c** Temporal variability in the *per capita* interaction effect measured by the SDs of the S-map coefficients over the experimental periods in the control treatment. Diamonds and circles indicate the overall means and raw data per replicate per year, respectively. **d** Density-dependence in the *per capita* interaction effect (IDD). For each interaction pair, S-map coefficients at each time point (grey dots) were plotted against the recipient standardized density at the corresponding time point. Solid segments represent simple linear regression lines with 95% confidence intervals (grey shading). The direction of the regression slopes (i.e., negative or positive interaction density-dependence (IDD)) is indicated by background colours: red, negative IDD; and blue, positive IDD. For this colouration, we did not consider the statistical significance of the regressions. The abbreviations of the community members are as follows: Phyt phytoplankton, Roti rotifers, C.zoop crustacean zooplankton, Macr macrophytes, Detr detritivores, Herb herbivores, P.pred phytophilous predators, B.pred benthic predators, N.pred neustonic predators, and Moll molluscs.

showed only subtle variability, but that on molluscs greatly varied over time (Fig. 3c).

By plotting the interaction effect against the recipient density at each time point, we found that there was considerable heterogeneity in the IDD (i.e., recipient density-dependence in interaction effect) among different interaction links in terms of not only its steepness but also its direction. The number of negative slopes (potentially stabilizing) was twice the number of positive slopes (potentially destabilizing) (Fig. 3d). Furthermore, density-independent interaction variability, represented by the deviation of the raw interaction effect data from the regression slopes in Fig. 3d, also differed considerably among different interaction links even when compared between similar slope steepnesses.

## Effects of interaction density-dependence on population sensitivity

We examined the effects of three types of interaction density-(in)dependence on recipient population sensitivity to the three pesticide treatments by using a multiple regression approach under the controls of mean interaction strength and recipient functions (Table 1, Fig. 4). Note that the effects of temporal variability shown in Table 1 and Fig. 4b, f, j were under the control of the dependence of interaction on recipient density, thereby representing the effects of the recipient-density-independent interaction variability. We observed that a greater magnitude of negative IDD tended to result in lower recipient sensitivity to pesticide disturbance (i.e., stabilizing effects) (Fig. 4a, e, i), whereas positive IDD was more likely to have neutral

**Table 1 | Relative importance of different interaction properties on recipient population sensitivity to the insecticide treatment (a, I vs. C), the herbicide treatment (b, H vs. C), and the insecticide + herbicide treatment (c, I + H vs. C)**

**(a) I vs. C**

| | likelihood ratio $\chi^2$ | df | P |
|---|---|---|---|
| Mean interaction strength | 0.28 | 1 | 0.59 |
| Interaction temporal variability | **26.78** | **1** | **<0.001** |
| Interaction density-dependence (IDD) | 0.35 | 1 | 0.55 |
| Direction of IDD (negative or positive) | 0.03 | 1 | 0.86 |
| IDD × direction of IDD | 0.41 | 1 | 0.52 |
| Recipient function | 2.88 | 2 | 0.24 |

**(b) H vs. C**

| | likelihood ratio $\chi^2$ | df | P |
|---|---|---|---|
| Mean interaction strength | 0.25 | 1 | 0.62 |
| Interaction temporal variability | **7.62** | **1** | **0.01** |
| Interaction density-dependence (IDD) | 0.68 | 1 | 0.41 |
| Direction of IDD (negative or positive) | 0.07 | 1 | 0.8 |
| IDD × direction of IDD | 1.06 | 1 | 0.3 |
| Recipient function | 1.23 | 2 | 0.54 |

**(c) I + H vs. C**

| | likelihood ratio $\chi^2$ | df | P |
|---|---|---|---|
| Mean interaction strength | 0.01 | 1 | 0.92 |
| Interaction temporal variability | **21.9** | **1** | **<0.001** |
| Interaction density-dependence (IDD) | 1.61 | 1 | 0.21 |
| Direction of IDD (negative or positive) | 0.12 | 1 | 0.73 |
| IDD × direction of IDD | **7.5** | **1** | **0.01** |
| Recipient function | **9.96** | **2** | **0.01** |

Likelihood ratio tests based on LMMs were conducted. Bold indicates statistical significance.

or positive effects on sensitivity (i.e., destabilizing effects) (Fig. 4a, e, i). The likelihood ratio $\chi^2$ of the interaction term between IDD × direction (negative or positive) of IDD was relatively high, although it was not statistically significant for the I and H treatments (Table 1). For every pesticide treatment, we observed positive effects of interaction temporal variability on recipient sensitivity to all pesticide treatments (Fig. 4b, f, j), suggesting that density-independent interaction variability was destabilizing. The interaction temporal variability had the greatest likelihood ratio $\chi^2$ (Table 1), suggesting that it was the most important variable for every treatment. Although weak interactions are considered stabilizing forces of communities, the mean interaction strength did not have a significant effect on recipient sensitivity to any pesticide treatment (Table 1, Fig. 4c, g, k).

## Discussion

Our study clearly demonstrated that different types of interaction density-(in)dependence certainly coexisted in the same community in the experimental paddies and that the interaction variability mediated the stability of the corresponding recipient populations (i.e., populations receiving interaction effects) embedded in the complex interaction webs. Consistent with our expectations, the different types of interaction variability had contrasting effects on population stability; negative IDD tended to decrease population sensitivity to pesticides (stabilizing), whereas positive IDD was more likely to magnify it (destabilizing). Previous studies have provided mixed predictions regarding the effects of interaction variability on population and/or community stability in response to external disturbances;

some studies suggested improving effects[24,31–34], while others suggested impairing effects[16,35]. In this context, we provided empirical evidence that whether interaction variability is stabilizing or destabilizing depends on the type of variability (i.e., how the *per capita* interaction effect responds to density changes), which was previously suggested only by theoretical works[23,36]. Furthermore, we found that not only positive IDD but also density-independent interaction variability consistently increased population sensitivity to pesticide treatments (destabilizing). We argue that incorporating the effects of density-independent interaction variability, or stochasticity in interaction strength (*sensu* Shoemaker et al.[37]), into the study of dynamic interaction networks is a promising avenue for future research.

The observed IDD was mostly biologically interpretable, especially in cases of seemingly prey–predator interactions. For example, the negative interaction effect of benthic predators (dragonfly larvae) and neustonic predators (water striders) on their potential prey molluscs became stronger with increasing density of molluscs (Fig. 3d; B.pred→Moll and N.pred→Moll), suggesting that these apex predators change their foraging behaviour in response to changes in prey density such that they choose more abundant prey. Such adaptive foraging in response to changes in prey density is regarded as a stabilizing force in predator–prey interactions[24,32–34] (but see Abrams[38]). On the other hand, the negative effect of phytophilous predators (damselfly larvae) on their potential prey (rotifers) was weaker with increasing rotifer density (Fig. 3d; P.pred→Roti), which was potentially destabilizing. Such density-dependence can arise from the saturating predation rate due to the constraints of predator handling time[21,39]. Unlike those interactions that were easily interpretable, there were interaction links whose density-dependence was hard to interpret. For example, the reasons why the negative interaction effect of phytoplankton on detritivores was negatively density dependent (Fig. 3d; Phyt→Detr) and why the positive interaction effect of rotifers on macrophytes was negatively density dependent (Fig. 3d; Roti→Macr) are unknown. Notably, the literature has shown that we still know little about functional forms of density-dependence in positive interaction effects such as the exploitation of benefits by a predator species from its prey[40] and mutualistic effects[41]. Our analysis sheds new light on never-detected IDD in real ecosystems, even though the specific underlying mechanisms are not yet clear. Identifying mechanisms may require further research on a theory that incorporates evolutionary processes in developing IDD[23].

Although there has been growing consensus that weak interactions play a significant role in community stability[4,7,42], in our study, we did not find any stabilizing effects of mean interaction strength (Fig. 4c, g, k). However, it is possible that interaction strength can influence population stability via the associations between mean interaction strength and interaction variability[43]. Berlow observed negative relationships between mean interaction strength and spatial variability, in which some weak interactions on average were extremely variable among spatial replicates, but strong interactions were less variable[44]. The author argued that the variability of weak interactions should play an important role in community and ecosystem organization. Our additional analysis revealed a similar pattern, an overall negative association between the mean interaction strength and interaction temporal variability (Supplementary Fig. 3a), although the trends were not statistically significant, probably due to data variance heterogeneity (Supplementary Fig. 3a). Interestingly, the pattern was mainly driven by negative density-dependent interactions, which were proven to be stabilizing (Supplementary Fig. 3b). As such, we suggest that future studies exploring the relationships between interaction strength and variability, including both density-dependent and density-independent interactions, are required to draw a complete picture of how those processes act interactively to determine community stability.

When interpreting our results, which were obtained from a combination of manipulative mesocosm experiments and EDM methods, some caution is needed. First, although there have been an increasing number of reports of EDM methods being applied in ecological studies[15,45–48], whether EDM can indeed uncover true causality from real data is still unclear[49,50] for

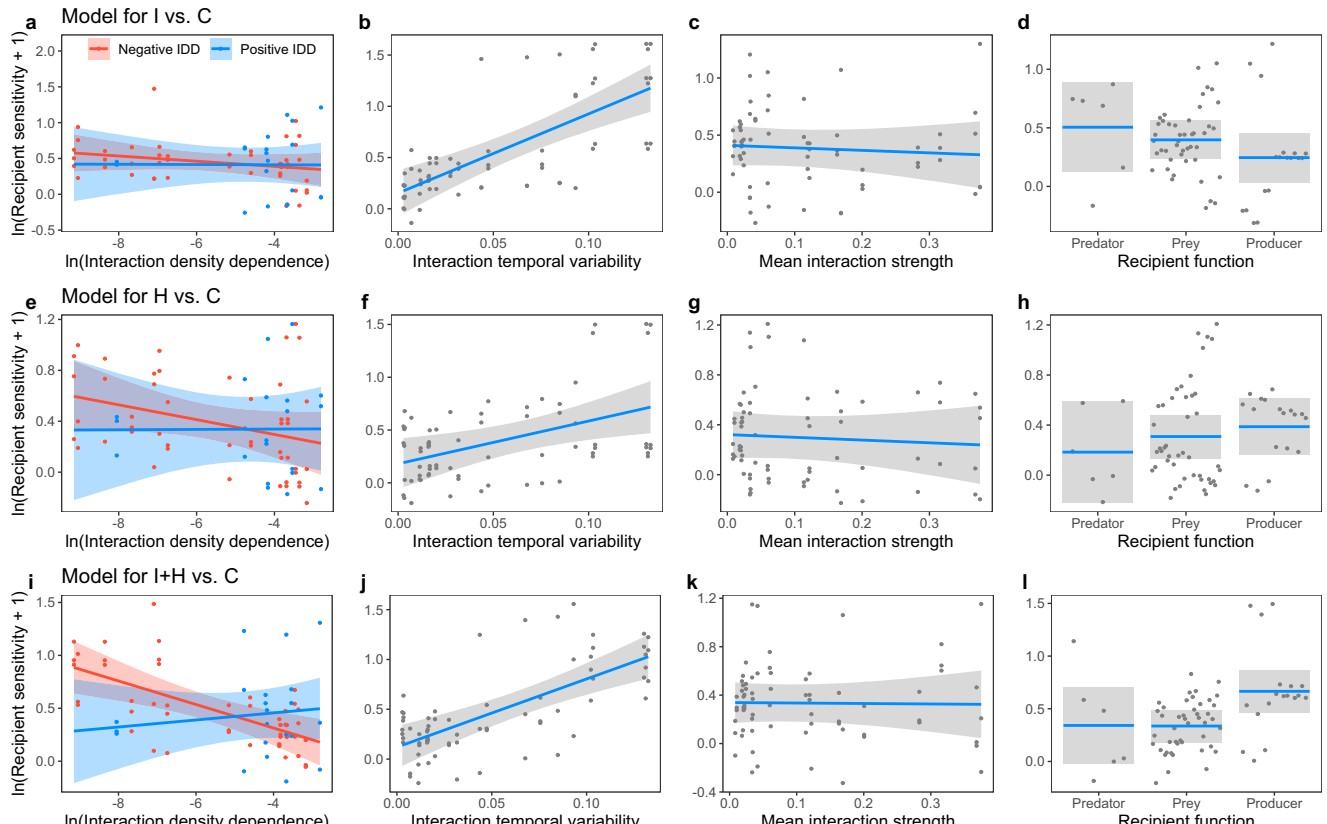

**Fig. 4 | Stabilizing and destabilizing effects of different interaction properties on recipient population sensitivity to the three pesticide treatments.** The insecticide treatment (I vs. C, **a–d**), herbicide treatment (H vs. C, **e–h**), and insecticide + herbicide treatment (I + H vs. C, **i–l**) are shown. Downwards (upwards) effects can be interpreted as stabilizing (destabilizing) for recipient density. The effects of interaction density-dependence (IDD) (**a, e, i**), interaction temporal variability (**b, f, j**), mean interaction strength (**c, g, k**), and recipient function (**d, h, l**) are shown. In each panel, fitted lines with 95% confidence intervals (grey shading) were obtained with all covariates held at median values. This indicates that the effects of the interaction temporal variability and mean interaction strength visualized in the figure are independent of density-dependent variability in interaction effects. In **a, e,** and **i**, the direction of IDD is shown in red (negative IDD) and blue (positive IDD). The grey, red, and blue dots are partial residuals. The number of dots ($n = 69$) is three times greater than the number of interaction pairs (23) because there are three years of replicates with a single interaction property value. A statistical summary of the models is shown in Table 1.

several reasons, such as violation of the assumptions of EDM on the stationarity of time series[26], synchronization among multiple time series due to seasonality, which are known to inhibit causal inferences by EDM[49], and missing variables (but see Wang et al.[51]). In our study, we performed a seasonal surrogate analysis (see Methods) to minimize the confounding effect of seasonality, although we recognize that the results of this analysis should be interpreted carefully[52,53]. Nevertheless, most of the interactions detected by CCM and quantified by S-map in our study were biologically reasonable (Fig. 3a, Supplementary Note 1).

Second, estimating interaction effects by S-map models has been considered sensitive to the choice of variables[15,54]. This becomes more serious when the number of interacting variables is large because, in such a case, potential combinations of variables become increasingly larger, leading to unstable quantification of interaction effects[55]. In our study, interaction effects were measured among the functional groups, not among the species. Although this simplification may be misleading because temporal species turnover within these groups could affect the observed interaction effects[56], it reduced the number of variables and hence the number of potential variable combinations. Thus, our estimation of interaction effects may be relatively robust. In addition, our observations were made over a two-week interval, and the interaction effects fluctuated over this relatively short time scale (Supplementary Fig. 2b). If such a time scale was actually shorter than the rate of temporal species turnover within functional groups, the effects of temporal changes in species composition within functional groups on the observed interaction variability may have been relatively minor.

Finally, the number of replicates used in our experiment was low (i.e., 2). Previous ecotoxicological, mesocosm studies have also used a relatively small number of replicates (ranging from 2–5), probably due to resource and/or effort limitations[28]. Further studies are needed to confirm that the observed interaction dynamics are reproducible. Nevertheless, our additional analysis confirmed that the main source of variation (in a statistical sense) in the interaction effect was census date and treatment, not residuals (Supplementary Fig. 4). Therefore, we believe that the temporal fluctuations of the interaction effect were relatively similar between the two replicates.

The ecological impacts of pesticides on multispecies communities include direct toxicity and indirect toxicity mediated by biotic interactions[1,28,57]. Our study emphasized the importance of the latter aspects of toxicity. Importantly, although the processes by which toxicity is mediated by interaction variability are apparently complex, some of the observed effects of the pesticides were consistent with previous studies. For example, the destructive effects of the insecticide fipronil on Odonata larvae (phytophilous and benthic predators in our study) and associated changes in community composition have been repeatedly reported in recent studies[27,28,58,59]. Community-level effects of the herbicide pentoxazone have not been well tested, but our study showed that negative effects of pentoxazone on macrophytes were consistently observed across the three experimental years. These results imply that even the causes and consequences of such established effects of these pesticides may involve density-(in)dependent interaction variability. This knowledge of the underlying mechanism is critical if we need to predict the impacts beyond the phenomenological understanding of them.

Recently, there has been a growing body of theoretical developments and new empirical studies on the species coexistence[60–63] and the stability of ecological communities without assuming equilibrium[15,64]. Understanding general laws about species diversity and stability in nonequilibrium communities would contribute to the understanding of the long-term effects of anthropogenic disturbances[65–67]. We suggest several promising directions for future studies. First, although field studies on nonequilibrium communities are not easy, they can be achieved by recently developed techniques such as the EDM framework. By combining this new technique with knowledge about the roles of interaction variability in determining community stability, we obtained a tool set to predict which communities are most sensitive to anthropogenic disturbances. Our study provides the first step in using the EDM framework effectively for such purposes. Second, scaling up from population stability to whole-community stability is needed to generalize our results. Our study examined population-level stability, not community-level stability. There are theoretical studies that have explored the role of density-dependence in *per capita* interaction effects on whole-community stability[23,36], but these studies did not account for density-independent interaction variability, which was found to be consistently destabilizing in our study. Furthermore, to our knowledge, there are no empirical studies on this topic. Scaling up to whole-community stability with full consideration of density-(in)dependent interactions would be a promising task for future studies. Compiling such studies could serve to improve novel ecological risk assessments, contributing to effective ecosystem management in the face of ongoing environmental changes due to human activities.

## Methods
### Experimental design
This study is a continuation of our previous study[28], which examined the single and combined effects of an insecticide and an herbicide on freshwater communities in paddy mesocosms. We chose fipronil and pentoxazone as test chemicals because both are commonly used in Japanese crop management, and their physicochemical properties are as typical as the properties of pesticides used in Japanese paddies[68]. Fipronil acts upon the central nervous systems of animals by inhibiting GABA receptors, which are dominant in the nervous systems of arthropods[69]. It is relatively stable in water and easily adsorbed to sediment, where it can persist for long periods[70]. Pentoxazone is an inhibitor of chlorophyll biosynthesis[71]. It is quickly degraded in water by hydrolysis but is readily adsorbed to sediment, where it can persist for a relatively long time[68,72]. Importantly, the direct toxicity of these chemicals is selective; fipronil is highly toxic to arthropods but moderately toxic to other animals and plants, while pentoxazone suppresses many vascular plants, but its toxicity to animals is low.

The experimental procedures used were described in Hashimoto et al.[28]. The experiment was conducted over the growing season of Japan (May-October), with air temperatures ranging between approx. 17 °C and 31 °C (mean daily temp.). In March 2017, we buried eight independent fibre-reinforced plastic (FRP) tanks (280 cm length × 120 cm width × 40 cm depth, RK 3014, KAISUIMAREN Co., Ltd., Toyama Prefecture, Japan) in the ground on the campus facilities of Kindai University (Nara Prefecture, Japan). Then, we spread sediments from uncontaminated areas near the study site on the bottom of each tank (to a depth of approximately 30 cm). Each tank was randomly assigned to one of four treatments, i.e., C: controls, I: insecticide alone, H: herbicide alone, and I + H: mixture of insecticide and herbicide.

Pesticide applications were performed at the beginning of the growing season of each experimental year (2017–2019), i.e., once a year. Specifically, we repeated the following procedure each year. In mid-April, the mesocosms were flooded with dechlorinated water to a depth of approximately 5 cm. From the end of May to early June, we transplanted insecticide-treated and control rice seedlings (Hino-hikari variety) via an array of 4 × 10 at 25 cm intervals. We applied an insecticide (fipronil) and an herbicide (pentoxazone) in the same way as recommended for commercial rice fields.

We treated nursery boxes of rice seedlings with Prince® (1% granular fipronil, HOKKO Chemical Industry, Inc., Tokyo, Japan) at a rate of 50 g/box 24 h prior to transplanting. Immediately after transplanting the rice seedlings, we applied Sainyoshi Flowable® (8.6% pentoxazone, KAKEN Pharmaceutical Co., Ltd., Tokyo, Japan) at a rate of 1.7 mL/tank (i.e., 500 mL/10-a) to the mesocosms. We confirmed that these pesticide applications succeeded by monitoring the temporal dynamics of the water and sediment concentrations of the insecticide and herbicide (Supplementary Fig. 5). The experiment was terminated in mid-October (i.e., approximately 140 days). We monitored the densities of ten functional groups in the community (eukaryotic phytoplankton, rotifers, crustacean zooplankton, macrophytes, detritivorous insects, herbivorous insects, phytophilous (clinging to macrophyte stems or leaves) predatory insects, benthic (living on bottom sediment) predatory insects, neustonic (living on the surface of water) predatory insects and molluscs; see Supplementary Table 1) every two weeks throughout the approximately 140-day experimental period in each experimental year. Aggregating species by functional groups has typically been adopted as a strategy for describing aquatic food webs[46,56,73,74]. As such, we acquired 8 tanks × 3 years = 24 fragments of time series, each of which had 10 consecutive timepoints, for each of the 10 community members. Preliminary sampling was done one week before and one week after the application of the pesticides, but the data were excluded to fulfil the requirement of the EDM analysis where the interval of time series must be constant.

Detailed monitoring methods are described below. To monitor the abundance of phytoplankton, rotifers, and crustacean zooplankton, we collected 500 mL water samples from 10 random sampling points in each experimental paddy. The samples were filtered through a 20-μm plankton net (Tanaka Sanjiro Co., Ltd., Fukuoka, Japan). We preserved the plankton in 5% acid Lugol's solution and then identified and counted them using an optical microscope (Olympus BX51, 100x, Olympus Corp., Tokyo, Japan) and a Sedgewick-Rafter counting chamber (Pyser-SGI Limited, Edenbridge, UK). Because most zooplankton cannot prey upon long-chained colonies of cyanobacteria, we used only eukaryotic phytoplankton data for phytoplankton counting. Note that we used different sampling protocols for crustacean zooplankton in 2017 and 2018–2019. In 2017, we sampled water (1 L) using the same method described above but filtered the samples through a 250-μm plankton net (RIGOSHA & Co., Ltd., Tokyo, Japan). Then, the zooplankton were preserved in 4% formalin and counted using a stereoscopic microscope (SMZ1500, Nikon Instech Co., Ltd., Tokyo, Japan). We monitored macrophyte density by setting three permanent quadrats (30 × 30 cm) in each mesocosm. We divided each quadrat into 36 grids (5 × 5 cm), counted the number of grids covered by a species and divided by the total number of grids (36). The total density of aquatic macrophytes was calculated as the sum of the coverage of each plant species. We collected aquatic macroinvertebrates by scooping a fishnet (1 mm mesh size) between the edges of each tank and rice seedlings (along a permanent transect). To reduce potential sampling errors and/or bias, fishnet scooping was conducted by the same person (Yuji Eguchi) throughout the experiment. We preserved specimens in 70% ethanol and then identified the species or closest taxonomic level and counted them.

### Statistics and reproducibility
**General information on the statistical analyses of the experimental data.** We obtained time series data from 8 independent mesocosm tanks (2 replicates for each treatment) 3 times (i.e., 3 years) and thus got 24 time series in total for each community member. Repeated measures were accounted for by including tank identity as a random variable when necessary (see the subsection 'Pesticide impacts on community member density'). All the statistical analyses were performed using the statistical environment 'R' Version 3.6.3[75]. All the statistical results can be reproduced by running R code deposited in https://doi.org/10.5281/zenodo.13609190[76].

**Data processing for EDM analysis.** Before all the analyses, the time series of density data were normalized by $\ln(x + 1)$ transformation (except for macrophytes) and then standardized to zero means and unit variances to facilitate comparisons among different taxa. Standardization was performed for all 24 fragments of time series data as a composite. Such data processing has been recommended in several EDM protocols[25,77,78].

**Pesticide impacts on community member density.** We examined the effects of pesticide application on the density of each member by using linear mixed models (LMMs) with the package 'glmmTMB' Version 1.0.2.1[79]. LMMs included standardized density as a response variable and treatment and census week (categorical) as explanatory variables. The random parts of the models were as follows: we specified tank identity and year as random intercepts and assumed an AR-1 temporal correlation structure within a given tank within a year. The statistical significance of the fixed terms was tested by Type III likelihood ratio tests, followed by Dunnett-type post hoc pairwise comparisons (two-sided) with the package 'emmeans' Version 1.5.2.1[80].

**Calculation of population sensitivity to pesticides.** To evaluate population sensitivity to the three pesticide treatments (as a proxy of population stability in response to pesticide disturbances), we used absolute values of the log response ratio (LRR). The LRR quantifies the proportional effects of experimental treatments on population density on a natural logarithmic scale, with positive values indicating greater population density in the treatment tanks and negative values indicating the opposite. We calculated $LRR_{i,j}$ for community member $i$ in year $j$ for every pesticide treatment as $\ln((T_{i,j} + 0.1)/(C_{i,j} + 0.1))$, where $T_{i,j}$ is the mean raw density (not standardized density) in either I, H, or I + H treatment and $C_{i,j}$ is the mean raw density of the controls. We added 0.1 to the numerator and denominator because there were several zero data points for both $T_{i,j}$ and $C_{i,j}$[30]. We used the absolute values of the LRR as the population sensitivity because our objective was to evaluate the stability of the populations rather than to determine whether the population increased or decreased in response to the treatments.

**Empirical dynamic modelling.** To evaluate the effects of biotic interactions within the communities in the experimental paddies, we performed EDM analyses. The essence of EDM is a reconstruction of the true dynamics of a system by a subset of the variables of that system, which is called state-space reconstruction or attractor reconstruction. This approach is ensured by the application of Takens' embedding theorem[81], which states that the true dynamics of a given system can be reconstructed by a set of time-lagged coordinates from a single variable of the system. Note that this theorem can be extended to cases in which there are multiple, not singular, observable variables[82]. That is, we can reconstruct an attractor based on a subset containing several variables of a system. Collectively, by using the EDM framework, we can draw inferences about the true dynamics of a given system by observing only a subset of variables from that system, even if that system comprises many variables, such as complex biological communities.

EDM was originally designed to analyse single, relatively long time series (e.g., at least >35–40 consecutive time points[77]), yet several alternative approaches have been suggested for analysing shorter time series by combining multiple time series that are too short to be analysed individually[54,83,84]. Moreover, such a method can be applied to analyse multiple short time series belonging to ecologically different conditions (i.e., different species and/or experimental treatments) if the dynamics of short time series with different conditions are sufficiently similar[83,85] (see Supplementary Note 2). Following these approaches, we assembled time series from the different experimental paddies and different years, which consisted of 10 time points × 8 paddies × 3 years = 240 time points in total, and analysed these assembled time series as a whole for each of our EDM analyses. Specifically, we allowed a single reconstructed attractor to consist of the whole data of every replicated tank, treatment and year, while we constrained every single data point on the attractor to be generated only from the data of the same replicate in the same year. This approach implicitly assumes that the time series of different replicates, treatments and years share dynamics that are sufficiently similar to analyse them together (Supplementary Note 2).

We first detected dynamic causality by performing convergent cross-mapping (CCM)[77] to identify interacting pairs of community members and determine their interaction directions. Then, we constructed a multivariate S-map[78] to track the time-varying, *per capita* interaction effect among interacting pairs determined by CCM. All EDM analyses were performed by using the package 'rEDM' Version 0.7.5[86].

**Detection of causalities by CCM.** CCM is a recently developed analysis used to test causalities between two variables that potentially interact with each other. In short, CCM examines the correspondence among reconstructed attractor manifolds to test causalities among variables. We tested the causalities of all pairs of 10 community members (Supplementary Tables 1, 3).

To prepare for performing the CCM and multivariate S-map analyses, we first determined the optimal embedding dimension $E$ of each community member, which is the embedding dimension that shows the best performance of the EDM predictions. The embedding dimension is the number of (time-lagged) coordinates used for state-space reconstruction. Specifically, we performed the simplex projection, which is one of the EDM methods for near-future predictions, for the whole time series of each community member, which is a composite of the data of all the tanks and every year, repeatedly by using embedding dimensions from 2 to 6. When reconstructing attractors, we concatenated fragments of time series following a previously suggested approach[84,85] (in detail, see the paragraph below for the CCM method). Then, we compared the prediction skills of the simplex projection among different embedding dimensions. Leave-one-out cross-validation (LOOCV) was used to determine the prediction skill. The prediction skill was judged by the Pearson correlation coefficient ($\rho$), mean absolute error (MAE) and root mean squared error (RMSE). These three criteria were consistent in most cases, but if they were not consistent, we used the criterion that graphically showed the most obvious peak form.

Next, to determine if one community member had a dynamic causality on another, we performed a CCM analysis[77]. As mentioned above, Takens' theorem states that attractor manifolds reconstructed by appropriate embedding preserve topological characteristics of the attractor of the whole system to which a variable used by the embedding belongs. As such, one can test whether two variables belong to the same dynamical system by comparing the topology of the reconstructed attractor manifolds of those variables. Furthermore, if variable A has a dynamic causation on variable B, a reconstructed attractor of variable B has complete information about variable A, but the reverse is not true. Thus, by examining the correspondence of the topology of the reconstructed attractors between variables A and B, we can infer not only whether there is causality between these variables but also whether the causal direction is from variables A to B, B to A, or bidirectional. Applying this principle, CCM examines the correspondence among reconstructed attractor manifolds to test causalities among variables.

We tested the causalities of all pairs of 10 community members (eukaryotic phytoplankton, rotifers, crustacean zooplankton, macrophytes, detritivorous insects, herbivorous insects, phytophilous predatory insects, benthic predatory insects, neustonic predatory insects, and molluscs). Specifically, first, we reconstructed an attractor manifold of each organism by using its best embedding dimension $E$. Although our time series had only 10 consecutive timepoints, state-space reconstruction sufficiently informative to perform CCM can be achieved by the ensemble of time series from different paddies and different years; thus, our analysis was performed with 10 timepoints × 8 paddies × 3 years = 240 timepoints in total (see Hsieh et al. and Clark et al.[83,84]). A total of 232 timepoints were available for macrophytes because we monitored the macrophytes 9 times, from the 4th week until the 20th week (not 10 times) in 2019. When reconstructing attractors, we

allowed one reconstructed attractor to consist of all the data for every replicate and year, while we constrained every single data point on the attractor to be generated only from the data of the same replicate in the same year. Second, we asked whether the reconstructed attractor manifold of a given community member (library member) could predict the value of another community member (target member). This method tested whether the former organism was causally affected by the latter. When performing predictions, we changed the amount of information of the reconstructed attractor ('library length' hereafter) from small to large by randomly sampling from the attractor, and we observed changes in the prediction performance ('cross-map skills' hereafter) with increasing library length. The minimum and maximum library lengths were $E + 1$ and $24 \times \{10 - (E - 1)\}$, respectively. The number of samples was 1000. When there was a causal effect from the target organism to the library organism, we could observe a monotonic increase in cross-map skills from small to large library lengths (i.e., 'convergence'). We determined the cross-map skills by the Pearson correlation coefficient ($\rho$). Note that assembling multiple short time series to reconstruct an attractor assumes that different fragments of time series share similar dynamic rules, which seemingly ignores the potential of differences in environmental conditions among experimental pesticide treatments. However, testing convergence requires a relatively large library length (because the relationships between cross-map skills and library length often show unimodal shapes, which cannot be distinguished with monotonic trends when the library length is too small), and in our case, such a large 'library length' was not achievable when we performed the analysis for each treatment separately.

We judged the causal effects of organism A on B when the CCM results met three criteria: (1) the difference in cross-map skills $\rho$ between the minimum and maximum library lengths ($\Delta\rho$) was greater than 0.1[15], (2) the cross-map skill at the maximum library length was significantly greater than that under the null hypothesis drawn from simulated surrogate data ($\alpha = 0.05$, although cases of $0.05 < P < 0.1$ were also considered in this study), and (3) the optimal time lag determined by lagged CCM[87] was not greater than 0. We tested these criteria sequentially in the order in which they are listed and proceeded to the next criterion only if the prior criterion was met.

There are several ways to simulate surrogate time series. Among these methods, we chose the 'seasonal surrogate' method, which is designed for time series with a constant cycle, such as seasons[88]. In short, this method detects and preserves a cyclic trend from an observed time series while randomly shuffling anomalies from the average of this cyclic trend. This gave a 'seasonal surrogate' time series that had the same cyclic trend relative to the original time series but with random anomalies. To simulate surrogate data, first, we treated 24 fragments of time series as one consecutive time series with a 10-timepoint cycle. Second, we generated surrogate data by using the function 'surrogate_seasonal' in the package 'rEDM' Version 0.7.5[86]. We simulated 1,000 surrogate data points for each of the 10 organisms. Then, we drew the distribution of cross-map skills under the null hypothesis (i.e., there was no causality) and calculated the $P$ values of the observed cross-map skills.

To determine the optimal time lag for cross-mapping, we performed lagged CCM[87]. Lagged CCM alters the time lag between the library and target variables when performing predictions, and one can compare cross-map skills among different time lags. Interestingly, lagged CCM can be applied to measure cross-map skills on the future state of target variables (i.e., time lags are positive). The better cross-map skills of a positive time lag than those of the current or past states of target variables can be regarded as a signal of false positives. In the lagged CCM analysis, we simulated 100 random samples for each time lag from $-2$ to $+2$ and compared the mean values of the cross-map skills $\rho$ among the time lags.

**Tracking interaction variability via multivariate S-map**. To estimate the *per capita* interaction effect among community members in the experimental paddies at each time point, we performed a multivariate S-map. The multivariate S-map is a locally weighted sequential linear

regression at each location of an attractor manifold reconstructed by multivariate embedding. Deyle et al. proposed that regression coefficients of the S-map ('S-map coefficients') at each time point can be interpreted as time-varying interaction effects[78]. That is, S-map coefficients are estimated at each time point of the time series data sequentially, yielding different values of interaction strength among different time points (i.e., 'time-varying'). This feature of S-map coefficients can be used to explore the context dependency of interaction effects[15,89]. For an S-map model to predict community member A, the estimated S-map coefficients of community member B are regarded as the time-varying interaction effect of community member B on A. To determine the interaction effect at the *per capita* level, we took the *per capita* population growth rate as the response variable[90]. The S-map coefficients estimated in this way theoretically correspond to the first-order partial derivatives of a recipient's *per capita* growth rate on a donor density: $\partial(1/N_r \times dN_r/dt)/\partial N_d$, where $N_r$ and $N_d$ denote the density of the focal recipient and donor, respectively[18,19]. The *per capita* growth rate was approximated by calculating $\ln(N_{t+1}/N_t)$, where $N_t$ and $N_{t+1}$ are the raw density data of the focal species at time points $t$ and $t + 1$, respectively. Since there were several zero values in the raw density data, one was added to every density data point to calculate the *per capita* growth rate. Note that S-map is sensitive to data with stochastic noise, and we used a modified version of S-map, namely, regularized S-map, which incorporates a penalty term to avoid overfitting[91]. Regularized S-map has been shown to provide relatively robust estimations of interaction effects[91]. As S-map analyses were performed for reconstructed manifolds embedded by using the whole data of eight paddies and three years as a composite (i.e., 240 time points), we obtained interaction effects not only at every time point but also for every paddy (and thus every treatment) and every year (Supplementary Fig. 2).

First, we chose multivariate coordinates to reconstruct a multivariate attractor manifold for each recipient (i.e., causally influenced by donors) by using the community members, the causalities of which were detected by CCM. Note that the values of the S-map coefficients were sensitive to the choice of embedding elements. There are several ways to reconstruct a multivariate attractor manifold. We adopted the approach in which the elements of univariate lagged embedding of a recipient were substituted by the donor densities to ensure that the dimension of multivariate embedding was no less than $E$. For example, when the $E$ of a recipient was 5 and when three donors were detected by CCM, multivariate embedding was performed by taking the coordinates $\{N_{Rt}, N_{Rt-1}, N_{D1t}, N_{D2t}, N_{D3t}\}$, where $N_{Rt}$ is the density of the recipient at time $t$ and $N_{Dit}$ is the density of donor $i$ ($i = 1, 2, 3$). We used the ensemble of time series from different paddies and different years to reconstruct multivariate attractors (the same as for simplex projection and CCM).

Second, we determined the optimal values of the two parameters ($\theta$ and $\lambda$) in S-map models by repeatedly performing multivariate S-map for each recipient. The parameter $\theta$ corresponds to the amount of weighting for locally weighted sequential regressions in the S-map, determining how locally the information from the whole state space is to be used for each regression. Thus, the nonlinearity of the reconstructed dynamics can be measured by $\theta$. When $\theta = 0$, the S-map becomes a nonweighted linear regression. The parameter $\lambda$ controls the L2 penalization to avoid overfitting[91]. We repeatedly performed a multivariate S-map to predict the *per capita* growth of recipient $\ln(N_{Rt+1}/N_{Rt})$ by changing the $\theta$ value by (0, 0.1, 0.5, 1, 1.5, 2, 2.5, 3, 4, 6, 8) and the $\lambda$ value by (0, 0.0001, 0.001, 0.01, 0.1, 0.5, 1, 2) for each multivariate embedding. Then, we compared the prediction skills (RMSEs) among different values of $\theta$ and $\lambda$ determined by LOOCV and used the values showing the minimum value of the RMSE for further analyses. Third, using the optimal $\theta$ and $\lambda$ values determined in the previous step, we again performed a multivariate S-map for each multivariate embedding. Optimal parameter values were reported in Supplementary Table 4.

Additionally, to confirm whether the temporal fluctuations of the interaction effect were consistent among replicates, we performed two-

way ANOVA for every interaction link, examining the effects of treatment, census date and their interaction on S-map coefficients. For simplicity, we did not account for any random effects or temporal autocorrelation.

**Effects of interaction density-dependence on population sensitivity.**
We performed multiple regressions to determine the effects of the interaction density-dependence (IDD) of each interaction link on the sensitivity of the corresponding recipient population. For this purpose, we first calculated the three interaction properties, IDD, interaction temporal variability and mean interaction strength.

IDD was represented by the absolute values of regression slopes between S-map coefficients and the standardized density of recipient populations. To do this, simple linear regressions using all treatment data over the three-year experimental period were performed for each interaction link (see Fig. 3d), and the absolute values of the regression coefficients for all the interaction links were obtained. Regardless of whether the direction of the slopes was negative or positive, higher absolute values of regression coefficients indicate stronger density-dependence in interaction effects (i.e., more variable interactions). The mean interaction strength or interaction temporal variability was calculated by first computing the mean or standard deviation of the S-map coefficients of the controls over the experimental period per year per replicate and then averaging their absolute values over all the years and replicates.

We constructed linear mixed models (LMMs) with the package 'glmmTMB' Version 1.0.2.1[79]. The LMMs included $\ln(x + 1)$-transformed absolute values of LRR (population sensitivity) as a response variable and all three interaction properties and recipient function (predator, prey, or producer) as explanatory variables. The IDD was $\ln(x)$-transformed to improve the model fit. In addition, to test whether the effects of IDD on the sensitivity of recipients were dependent on the direction of IDD, we included the direction of IDD (negative or positive) and IDD × direction of IDD interaction in the above LMMs as explanatory variables. The identities of the interacting pairs of community members and year were specified as random intercepts. Models were generated for all three pesticide treatments. The importance of each explanatory variable was evaluated by the Type III likelihood ratio $\chi^2$ computed by the package 'afex' Version 0.28.0[92].

## Reporting summary
Further information on research design is available in the Nature Portfolio Reporting Summary linked to this article.

## Data availability
All the data including time-series data of freshwater communities and pesticide concentration in the paddy mesocosms are freely available at: https://github.com/KoyaHashimoto/PaddyInteractionVariability (https://doi.org/10.5281/zenodo.13609190[76]).

## Code availability
The R code used in this manuscript is freely available at: https://github.com/KoyaHashimoto/PaddyInteractionVariability (https://doi.org/10.5281/zenodo.13609190[76]).

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

## Acknowledgements

Dr. Masayuki Ushio (HKUST) assisted greatly with our EDM analyses. We are particularly grateful to the members of the Laboratory for Conservation Ecology of Kindai University for helping with our field experiment and survey. We also thank Dr. Robin J. Smith (Lake Biwa Museum) for identifying zooplankton crustaceans, especially ostracods. The authors wish to thank Dr. Tadao Kitagawa and Dr. Takuo Sawahata (Kindai Univ.) for valuable technical advice. The members of the Biodiversity Assessment and Projection Section and the Climate Change Impacts Assessment Research Section of the NIES greatly helped us by joining discussions. The present study was supported by the Environment Research and Technology Development Fund (ERTDF) FY2017 (4-1701) of the Ministry of the Environment, Japan, the Japan Society for the Promotion of Science (JSPS) KAKENHI (Grant Numbers 20K15640 and 21J01194 to K. Hashimoto and 21K18318 to D. Hayasaka and T. Kadoya), and BRIDGE FY2024 (R6-09) of the Cabinet Office, Japan.

## Author contributions

K.H., D.H., and T.K. conceived and designed the study. Y.E., Y.S., J.C., and K.H. collected the data. K.H. analysed the results and prepared the figures and tables. K.H., D.H., and T.K. led the writing of the manuscript. K.H., D.H., Y.E., Y.S., J.C., K.S., K.G., and T.K. contributed to revising the earlier draft. All authors have given final approval to submit this manuscript and agree to be accountable for the aspects of the work that they conducted.

## Competing interests

The authors declare no competing interests.
