## [Transparent Peer Review file · Communications Biology]

Multifaceted effects of variable biotic interactions on population stability in complex interaction webs

Corresponding Author: Dr Koya Hashimoto

Version 0:

Reviewer comments:

Reviewer #1

(Remarks to the Author)

This study investigated the impacts of pesticides on the aquatic communities in the paddy field that is a dominant landuse type in agricultural systems in east Asia. The strength of this study is the solid application of empirical dynamic modeling (EDM) on time series of population dynamics of 10 groups of organisms and demonstrate the linkage between the disturbance by pesticides, changes in interaction strength, and degree of population size responses. Most interesting discovery is the unimodal relationship between the degree of interaction strength changes and population density changes by disturbances, which clearly demonstrated the nonmonotonic response of biological communities against anthropogenic disturbance. General strategies of statistical analyses and EDM are quite reasonable and extensive so the results are convincing. So this new discovery should be widely shared by community ecologists as well as environmental managers. However, at the same time, I found two major issues; one is about the unclarity and less convincing methodological details in the analyses and the other is about the storyline in Introduction and Discussion. These two issues should be addressed and considered for better presentation of this study.

Issues in analyses

1) In order to solve a common issue of shortage of time series length of ecological datasets, the authors reasonably combined multiple time series for effectively applying EDM methods. However, the actual length of the combined time series for each piece of EDM analyses is not well presented. For each of 10 groups of organisms (10 community members), as the authors pointed out, there are $8 \times 3 \times 10 = 240$ time points, but it includes the time series under four distinct treatments. When applying each of EDM methods (simplex projection, CCM, and multivariate S-map), how did the authors separate these 240 time points and why? For example, based on the results shown in Fig. 2a, it seems that the authors reconstructed interaction strength for each treatment separately so the time series length for multivariate S-map is $240/4 = 60$ or $240/(4^2) = 30$ (for each replicate independently)? This should be more clearly mentioned for ensuring reproducibility of the results. In addition, if the authors treated each replicate for each treatment independently in the EDM analysis, the implicit assumption would be that the dynamics and its attractor was different between replicates and treatments. The authors would want to explicitly mention such assumption and justify it. These descriptions would be necessary for other steps of EDM analyses.

2) Line 383-: Although it is reasonable to standardize the population size to zero means and unit variances for the authors' objectives, it is not clear how the 240 time points for each community member was used for this standardization. Did the authors standardize the mean and variance for the whole 240 time points, or for subsets of them?

3) Line 402-: Speculated from relatively narrow value range of the y-axis of Fig.4, the density changes due to the pesticides were calculated based on the mean and variance standardized datasets. The authors would want to make the calculation method more explicitly and justify it. To me, since the authors used this value as a proxy of stability of populations against pesticide disturbance, it would be more reasonable to use the raw value of population sizes and also used the per-capita-based index, e.g. the ratio of the mean (raw) density in the treatment with pesticides to those in the control, rather than, the absolute value of their subtraction.

4) I am not fully convinced by the direct usage of the S-map coefficient as the interaction strength (i.e., the per-capita impact of the donor community member on the population growth rate of the recipient community member). As another method is also recognized by the authors and mentioned in the text, I prefer to use the per-capita impact of the donor community member on the "per-capita" population growth rate of the recipient community member, which can be calculated by the division of the S-map coefficient by the recipient population size. The authors justified not using the per-capita interaction

strength by the argument that the "true" interspecific interactions can not be necessarily written by generalized LV formulations (in line 187 in supplement). I do not agree with this justification; even if it is not possible to assume specific mathematical formulations for interspecific interactions, independently of specific formulations, per-capita population growth rate of the recipient species has a robust ecological meaning to indicate the potential of population growth (or decline). Note that in evolutionary ecology and community ecology, the per-capita growth rate of a focal population has been used as important indices for checking the invasibility of rare species and index to check the coexistence of multiple species in non-equilibrium state.

Although I do NOT intend to insist that the per-capita interaction strength better fit to the goals of this study, the authors need another justification for directly using the S-map coefficients for IS. In fact, this is also related to a part of issues in Introduction and Discussion (see below).

5) I would also point out a technical issue of the direct usage of the S-map coefficients as IS. As is shown in Fig.4, it seems that the authors would intend to link the temporal variability of IS to the density changes of the recipient population. Here, please note that the temporal changes (and responses to disturbances) of IS include the changes in the population size of the recipient population. It implies the risk of statistical artifact of the relationship between IS changes and density changes. I hope this is my misunderstanding but the authors would want to make clearer this point.

minor point

Line 447-448: "real time" --what does it mean? Would it be "every time"?

Issues in storyline and arguments in Introduction and Discussion

1) With the usage of the S-map coefficients as the index of IS, I am not fully convinced by (i) the criticism on a 'static view' of interaction networks and classical mathematical ecology papers (e.g. May 1972) in Introduction (line 76-), and (ii) the argument on the possible mechanisms to realized temporal variability of IS (from Line 246), which addresses the flexible foraging. Including myself, I am afraid that not a few scientists who apply EDM to estimate interaction strength oversell the temporal variability of S-map coefficients as an evidence of temporal variability of interaction strengths. Even with the 'static view' in the classical mathematical formulations, which use the temporally-constant interaction coefficients, the S-map coefficients changes with time under non-equilibrium state since the population size of the recipient species continuously changes with time. Thus I feel that the authors need to choose either of two alternative solutions; using the per-capita IS for the whole analyses with keeping the storyline or keeping using the current IS but dramatically softening the criticism (i) and the argument (ii) in the text.

2) Although I am sure that the results shown in Fig.4a-c have high novelty, I am not convinced at all by the authors' argument to link this "non-monotonic patterns" to "nonlinear response" of communities against disturbance and to the shift between buffering and amplifying effects by variabilities of IS. First, it is not acceptable to loosely use the term "nonlinear" or "nonlinearity" when just encountering nonlinear mathematical formulations. Please note that most of differential equation models and difference equation models of population dynamics use the nonlinear mathematical formulations to represent population growth and interspecific interactions but it does not imply the population dynamics governed by such "nonlinear" equations are nonlinear. Temporal fluctuations around the locally asymptotically stable equilibrium and those around the locally asymptotically stable limit cycles are evaluated as "linear" when applying univariate S-map; only when the attractors of the dynamics is torus or pseudo-strange attractors, the univariate S-map regards it as the nonlinear dynamics. It is confusing to use the term "nonlinear" in two distinct meanings in a manuscript that uses EDM analyses. I would say this is just a minor comment but more importantly, I do not think the non-monotonic patterns in Fig. 4a-c are related to the buffering nor amplification effects. When one wants to check the presence of buffering or amplifying effects of a focal process (changes of IS in this study), it is necessary to compare the responses (density changes in this study) with the focal process and those without focal process as the baseline. However, in this study, since it would be difficult to predict or extrapolate the density responses without IS changes, I am afraid that the authors' argument is highly speculative and even misleading. It would be nicer if the authors could present more direct interpretation of this pattern in Discussion as well as in abstract.

Reviewer #2

(Remarks to the Author)

In this study, the authors explored whether the temporal flexibility of species interaction mitigates or amplifies the effect of anthropogenic disturbances on population dynamics. The detail can be summarized as follows.

Specifically, applying the EDM (empirical dynamic modelling) approach to the time series dataset of multitrophic mesocosms under four disturbance conditions (control, insecticide, herbicide and their combination), they quantified 1) the impact of disturbances on the density of the community members, 2) IS responses: changes in the magnitude of species interaction caused by disturbances and 3) IS intrinsic variability: temporal variability of species interaction under no disturbances. As a result, they found the nonlinear, concave relationship between IS responses and the impact of disturbances on the recipient density and the indirect path from IS intrinsic variability to the impact of disturbances via the IS responses. Based on these findings, they concluded that the effect of species interaction on the impact of anthropogenic disturbances depends on the interacting pairs but may be predictable by the information of the IS intrinsic variability.

Overall, the paper is well written: the text is easy to follow, and the methods/results are adequately described to be reproducible. I also agree with the ecological importance of the topic covered in this study. However, I have two concerns about the measure of interaction strengths used in this study.

First, the authors used the multivariate S-maps to quantify the strength of biological interactions; however, this method is known to be vulnerable to process noise, which includes the substantial variance in the estimated interaction strengths (Cenci et al. 2019). This point would be critical for this study: the indirect path from the variance of interaction strengths to the impact of disturbance (process noise) may be an artifact caused by the model's defect. Therefore, to remove this possibility, I suggest the authors reanalysed the data by use of the S-maps with regularization terms (a.k.a regularized S-map, Cenci et al. 2019).

The second concern may be more critical. Specifically, S-map methods sequentially measured the Jacobian elements, which is mathematically identical to the partial derivative of the response variable with respect to the explanatory variable. Thus, if the raw values of population density are used as the model variables, the S-map coefficients corresponds to elements of the well-known 'community matrix' in ecological theory, which is the form as $X_i(t) \cdot \partial F_i / \partial X_j(t)$, in which $X_i(t)$, $X_j(t)$ and F_i stands for the density of recipient species at time t, that of donor species and the unknown function of per capita population growth of species i, respectively. As you see, the S-map coefficients obtained in this manner contain information on the density of the recipient species, making it difficult to use simple statistical modelling to analyze the relationship between the coefficients and the recipient's density. For example, it might be that IS responses did not mediate the impact of disturbances but simply that changes in the recipient's density were reflected in changes in the interaction strength. This defect cannot be resolved in the current S-map framework (Chang et al. 2021); however, the authors have to control information on the recipient's density, at least in their statistical analysis.

Reviewer #3

(Remarks to the Author)

The manuscript explores the effects of disturbance on interaction strength within mesocosm communities. They focus on pesticides as a disturbance, using both insecticide and herbicide, and show that variability in interaction strength can buffer the impacts of disturbance. However, this effect occurs only up to a certain disturbance threshold, beyond which variability in interaction strength amplifies the effects of disturbance. They combine experiments in mesocosm communities with empirical dynamic modeling to estimate interaction strength and understand the effect of disturbances in these communities.

One of the interesting aspects of this study is that in their experiments, they chose to use two different types of pesticides (insecticide and herbicide), which are common anthropogenic disturbances found in agricultural habitats (which is their interest). The authors find insecticide has a stronger effect on community members than herbicide. I do wonder whether this result is simply an effect of having more members in their mesocosm communities being susceptible to insecticide than herbicide. Meaning if most members are indeed insects or animals rather than plants or algae wouldn't the members from an animal kingdom be more directly affected by insecticide? This seems to be what they find, so it is not surprising but how much of their mesocosm is composed of insects?

One of the weaknesses I see with the manuscript is the authors' choice of monitoring community members as wide categories (functional groups). I understand they model effective interactions between community "guilds" but I do wonder how much they are missing in terms of interactions within each guild. Could some of the patterns that they observe in terms of variability in interaction strength be driven by dynamics happening within each category? How problematic is it to aggregate all herbivores (or other categories) of a given sub community and disregard all the interactions between members of the same category? Another point the authors briefly touch in their discussion is species turn over within each of these categories and how much of their results (including but not restricted to intrinsic variability) could be driven simply by species turnover. I am not sure I understand why fluctuations in interaction strength observed over short period of times rules out the effects of species turn over.

One of your key results depends on the severity of the disturbance, meaning that as disturbance gets more severe, the effect of variability in interaction strength changes from buffer to amplification. From an empirical perspective, I wonder if the authors have a suggestion on how to estimate when a disturbance becomes severe? From their methodological point of view, if I understood correctly they gradually increased disturbance until it reached the threshold for which it changed from buffering to amplification? Would they still find the same result if instead of gradually increasing pesticide if they had different levels of pesticides to begin with? Meaning, are their results the same whether the disturbance is a press or pulse one?

The authors report changes in interaction strength over the years even in the control treatment which they refer to as an intrinsic characteristic of the community. I believe there are a few points related to this finding that require further exploration. It is not clear to me how exactly the authors effectively separate the effects of this intrinsic variability from the variability found in their experimental treatments. I understand they only have two replicates of each treatment and control but I do wonder if the replicates in the control changed in a similar way? Would this intrinsic variability be an effect of abiotic/environmental effects? If that is the case (environmental driven) would they expect similar changes to occur in the control with no pesticide as well as the other treatments equally? It would be interesting to discuss whether there were groups within the community for which intrinsic variability was greater than for others (for instance whether insects within their communities showed greater intrinsic interaction variability as they were possibly also the ones more heavily affected by the insecticide disturbance).

Minor comments

Line 133: "disturbances become more severe" does this mean higher levels of pesticide? Is it incremental?

How consistent are the estimates of interaction strength across your replicates (not only control but pesticide treatments as well)?

Pesticide application (whether is gradually done or as a single application) needs further explanation. I understand this is a continuation of another study and that methods are fully explained there, but given that the severity of disturbance is a key point in their manuscript this needs to be clarified.

Version 1:

Reviewer comments:

Reviewer #1

(Remarks to the Author)

The revision substantially considered my comments from the previous round of the reviewing process. I am happy to see that the authors' appropriately switched the index of interaction strength toward per capita based and developed a new idea of interaction density dependent (IDD). I am convinced by the new results showing the linkage between the sign of IDD, magnitude of IDD, and the population sensitivities against pesticides. However, I still have a concern on technical aspects on their study. The following are the list of specific comments:

Major comments: I am sorry that all the below are linked each other and somehow redundant each other.

1) I am not fully convinced by the method combining all data together to reconstruct a single common attractor while assuming (claiming) that interaction properties (such as mean interaction strength) were different among treatments.

More naturally, the authors could use 2 paddies x 3 years x 10 censuses 60 time points per each treatment to reconstruct treatment-specific attractor.

Or, did the authors let each paddy experience different treatments year by year?

The authors might need to technically justify this point.

2) For a technical justification, only when the forecasting ability from a single attractor and those from treatment-specific attractor are not different, the authors' method would be fine. If the forecasting ability is higher in a single attractor, it implies that all treatments are under the identical dynamical system. It is apparently nice but how can the authors claim that the interaction properties were different between treatments on a single common attractor? If the forecasting ability is lower, it implies that multiple attractors are mixed and interfered in the reconstruction and that the time series from different treatments belong to different attractor. Then, why not using the treatment-specific attractors?

3) It is difficult for me to imagine the situation where the time series from different treatments tend to stay longer in treatment-specific parts of the common attractor, resulting in the situation in which distinct interaction networks with different interaction magnitude were realized (Fig. S2). It is much more straightforward to reconstruct treatment-specific attractors and compared the interaction properties, in order to demonstrate such treatment-specific interaction properties.

Minor comment:

Line 235: The statement is confusing. Why not using more direct statement, e.g. "greater magnitude of negative IDD resulted in smaller recipient sensitivity to pesticide disturbance"?

Figure 3d: The color (red or blue) implies that there were no interaction pairs with density independent IDD?

Fig.4: I could not fully understand the data structure even if I carefully read the method part. Based on the results shown in Fig.3d, the sign of IDD (positive or negative) and the magnitude of the IDD (by the absolute value of the regression slope) were calculated in 23 pairs of the interactions. On the other hand, there apparently exist more than 23 data points in Fig.4, e.g. panels a, e, and l. Is this because with a single interaction density dependence value, there are three years' replicates?

REVIEWERS' COMMENTS and AUTHORS' RESPONSES

Reviewer #1 (Remarks to the Author):

This study investigated the impacts of pesticides on the aquatic communities in the paddy field that is a dominant landuse type in agricultural systems in east Asia. The strength of this study is the solid application of empirical dynamic modeling (EDM) on time series of population dynamics of 10 groups of organisms and demonstrate the linkage between the disturbance by pesticides, changes in interaction strength, and degree of population size responses. Most interesting discovery is the unimodal relationship between the degree of interaction strength changes and population density changes by disturbances, which clearly demonstrated the nonmonotonic response of biological communities against anthropogenic disturbance. General strategies of statistical analyses and EDM are quite reasonable and extensive so the results are convincing. So this new discovery should be widely shared by community ecologists as well as environmental managers. However, at the same time, I found two major issues; one is about the unclarity and less convincing methodological details in the analyses and the other is about the storyline in Introduction and Discussion. These two issues should be addressed and considered for better presentation of this study.

Thank you for evaluating the potential of our manuscript. In particular, we thank you for your comments about our criticism of the 'static view' of community ecology, which led us to think more deeply about interaction variability. As such, we switched the use of interaction effects from the population level to the individual level. This substantially affects the results. In particular, the unimodal relationship between the degrees of interaction strength changes and population density changes caused by disturbances was replaced by our new results. However, we believe that our new results provide deeper insights into the stabilizing/destabilizing roles of interaction variability while allowing us to keep our critique against the 'static view' of interaction webs. The summary of the revision is described as follows.

1. We switched the use of model coefficients from population-level to individual-level coefficients. As such, the storyline regarding the complaint on the 'static view' of ecological communities was kept.
2. Using the *per capita* interaction effect allowed us to perform a deeper analysis. Specifically, we can describe the density-(in)dependence of the interaction effect and evaluate the effects of how interactions are density-dependent on population stability.
3. Following this approach, we set our goals as (1) describing various forms of interaction density-dependence in a real freshwater community and (2) evaluating the effects of different types of interaction density-dependence on population stability (resilience against pesticide

impacts). Therefore, the abstract, introduction, analyses, results and discussion were completely revised.

4. Furthermore, we developed three specific hypotheses regarding interaction density-(in)dependence: (1) negative interaction density-dependence (IDD) increases the stability of the recipient population (Fig. 1a), (2) positive IDD increases the instability of the recipient population (Fig. 1b), and (3) density-independent interaction variability is more likely to destabilize the recipient population (Fig. 1c).
5. Our revision of the results is as follows: (1) Reconstructed interaction networks by using the *per capita* interaction effect instead of the population-level interaction effect. (2) The IDD of each interaction link is described. (3) The effects of IDD (i.e., the effects of density-dependent interaction variability) on population instability were evaluated. Population instability was also evaluated at the *per capita* level using the absolute values of the log response ratio. All the other analyses regarding the interaction effect in the previous version were replaced as well.
6. Intriguingly, negative IDD tended to stabilize populations, whereas positive IDD tended to destabilize, which is consistent with a theoretical prediction proposed by previous studies (Kawatsu and Kondoh 2018; Kawatsu 2020). We speculate that the nonmonotonic, concave relationships between interaction change and population change observed in the previous version were generated by the mixture of the stabilizing and destabilizing effects of negative/positive IDDs.
7. In addition, we changed the title from “Severe disturbance overflows the stabilizing buffer of variable biotic interactions” to “Multifaceted effects of variable biotic interactions on population stability in complex interaction webs” to better represent the content of the revised version.

Kawatsu, K. (2020). Ecology and evolution of density-dependence. In: *Diversity of Functional Traits and Interactions: Perspectives on Community Dynamics* (ed. Mougi, A.). Springer, Singapore, pp. 161–174.

Kawatsu, K. & Kondoh, M. (2018). Density-dependent interspecific interactions and the complexity – stability relationship. *Proc. R. Soc. B Biol. Sci.*, 285, 20180698.

Issues in analyses

1) In order to solve a common issue of shortage of time series length of ecological datasets, the authors reasonably combined multiple time series for effectively applying EDM methods. However, the actual length of the combined time series for each piece of EDM analyses is not well presented. For each of 10 groups of organisms (10 community members), as the authors pointed out, there are 8

*x 3 x 10 = 240 time points, but it includes the time series under four distinct treatments. When applying each of EDM methods (simplex projection, CCM, and multivariate S-map), how did the authors separate these 240 time points and why? For example, based on the results shown in Fig. 2a, it seems that the authors reconstructed interaction strength for each treatment separately so the time series length for multivariate S-map is $240/4 = 60$ or $240/(4*2) = 30$ (for each replicate independently)? This should be more clearly mentioned for ensuring reproducibility of the results. In addition, if the authors treated each replicate for each treatment independently in the EDM analysis, the implicit assumption would be that the dynamics and its attractor was different between replicates and treatments. The authors would want to explicitly mention such assumption and justify it. These descriptions would be necessary for other steps of EDM analyses.*

We apologize for the unclear explanations. We used multiple time series of 8 paddies x 3 years x 10 censuses = 240 time points to reconstruct a single attractor when we applied all the EDM methods. Thus, the actual length of the time series was 240, and our analysis did not separate the time series into replicates or treatments. Specifically, we allowed for one reconstructed attractor to consist of the whole data of every replicate and year, while we constrained every single data point on the attractor to be generated only from the data of the same replicate in the same year. S-map coefficients were based on such concatenate reconstructed attractors; therefore, we obtained S-map coefficients not only for every time point but also for every paddy (and thus every treatment) and every year. In addition, in this way, we assumed that each replicate and treatment had a common rule of dynamics. We explicitly noted these points in the Methods section: “Specifically, we allowed a single reconstructed attractor to consist of the whole data of every replicate, treatment and year, while we constrained every single data point on the attractor to be generated only from the data of the same replicate in the same year. This approach implicitly assumes that the time series of different replicates, treatments and years share a common dynamic rule” (LL 499-503).

2) Line 383-: Although it is reasonable to standardize the population size to zero means and unit variances for the authors' objectives, it is not clear how the 240 time points for each community member was used for this standardization. Did the authors standardize the mean and variance for the whole 240 time points, or for subsets of them?

We standardized the mean and variance for all 240 time points. We explicitly explained this in the methods subsection “Data processing for the EDM analysis” as follows: “Before all the analyses, the time series of density data were normalized by $\ln(x + 1)$ transformation (except for macrophytes) and then standardized to zero means and unit variances to facilitate comparisons among different taxa.

Standardization was performed for all 24 fragments of time series data as a composite.” (LL 454-457).

3) Line 402-: *Specualted from relatively narrow value range of the y-axis of Fig.4, the density changes due to the pesticides were calculated based on the mean and variance standardized datasets. The authors would want to make the calculation method more explicitly and justify it. To me, since the authors used this value as an proxy of stability of populations against pesticide disturbance, it would be more reasonable to use the raw value of population sizes and and also used the per-capita-based index, e.g. the ratio of the mean (raw) density in the treatment with pesticides to those in the control, rather than, the absolute value of their subtraction.*

As the reviewer noted, we used density changes as a proxy for population stability. We changed the calculation of density changes by using a *per capita*-based index, i.e., the absolute value of the log response ratio of the raw value of population density in the pesticide treatment to that of the controls. This facilitates the interpretation of our results because this index is independent of the proportional effects of population density on pesticide use. In addition, we referred to this *per capita*-based index as population sensitivity to pesticide treatments.

We briefly explained this in the Results section as follows: “We evaluated population sensitivity to pesticide treatments, which is a proxy of population instability, by calculating the absolute value of the log response ratio (LRR) of the mean density in a pesticide treatment relative to that in the controls (Hedges *et al.* 1999). We calculated $LRR_{i,j}$ for community member i in year j for every pesticide treatment as $\ln((T_{i,j} + 0.1)/(C_{i,j} + 0.1))$, where $T_{i,j}$ is the mean density in either the I, H, or I+H treatment and $C_{i,j}$ is the mean density of the controls. We added 0.1 to the numerator and denominator because there were several zero data points for both $T_{i,j}$ and $C_{i,j}$ (Martinson & Raupp 2013).” (LL 183-190), which are explained in more detail in the methods section as follows: “To evaluate population sensitivity to the three pesticide treatments (as a proxy of population stability in response to pesticide disturbances), we used absolute values of the log response ratio (LRR). The LRR quantifies the proportional effects of experimental treatments on population density on a natural logarithmic scale, with positive values indicating greater population density in the treatment tanks and negative values indicating the opposite. We calculated $LRR_{i,j}$ for community member i in year j for every pesticide treatment as $\ln((T_{i,j} + 0.1)/(C_{i,j} + 0.1))$, where $T_{i,j}$ is the mean density in either I, H, or I+H treatment and $C_{i,j}$ is the mean density of the controls. We added 0.1 to the numerator and denominator because there were several zero data points for both $T_{i,j}$ and $C_{i,j}$ (Martinson & Raupp 2013). We used the absolute values of the LRR as the population sensitivity because our objective was to evaluate the stability of the populations rather than to determine

whether the population increased or decreased in response to the treatments” (LL 473-485).

4) I am not fully convinced by the direct usage of the S-map coefficient as the interaction strength (i.e., the per-capita impact of the donor community member on the population growth rate of the recipient community member). As the another method is also recognized by the authors and mentioned in the text, I prefer to use the per-capita impact of the donor community member on the "per-capita" population growth rate of the recipient community member, which can be calculated by the division of the S-map coefficient by the recipient population size. The authors justified not using the per-capita interaction strength by the argument that the "true" interspecific interactions can not be necessarily written by generalized LV formulations (in line 187 in supplement). I do not agree with this justification; even if it is not possible to assume specific mathematical formulations for interspecific interactions, independently of specific formulations, per-capita population growth rate of the recipient species has a robust ecological meaning to indicate the potential of population growth (or decline). Note that in evolutionary ecology and community ecology, the per-capita growth rate of a focal population has been used as important indices for checking the invasibility of rare species and index to check the coexistence of multiple species in non-equilibrium state. Although I do NOT intend to insist that the per-capita interaction strength better fit to the goals of this study, the authors need another justification for directly using the S-map coefficients for IS. In fact, this is also related to a part of issues in Introduction and Discussion (see below).

As suggested, we realized that irrespective of the specific mathematical formulation of community dynamics, the *per capita* population growth rate can indeed be calculated and has ecological relevance. Therefore, we switched the use of model coefficients from population-level to individual-level coefficients by changing the response variable of S-map from the population growth rate to the *per capita* growth rate. This corresponds to the formal definition of *per capita* interaction effects formulated by Travis & Post (1979) and Novak *et al.* (2016): the first-order partial derivatives of a recipient's *per capita* growth rate on a donor density, $\partial(1/N_r \times dN_r/dt)/\partial N_d$, where N_r and N_d denote the density of the focal recipient and donor, respectively.

We explained the detailed methods in the methods section as follows: “To determine the interaction effect at the *per capita* level, we took the *per capita* population growth rate as the response variable (Suzuki *et al.* 2017). The S-map coefficients estimated in this way theoretically correspond to the first-order partial derivatives of a recipient's *per capita* growth rate on a donor density: $\partial(1/N_r \times dN_r/dt)/\partial N_d$, where N_r and N_d denote the density of the focal recipient and donor, respectively (Travis & Post 1979; Novak *et al.* 2016). The *per capita* growth rate was approximated by calculating $\ln(N_{t+1}/N_t)$, where N_t and N_{t+1} are the raw density data of the focal species at time

points t and $t+1$, respectively. Since there were several zero values in the raw density data, one was added to every density data point to calculate the *per capita* growth rate” (LL 526-536).

Novak, M., Yeakel, J.D., Noble, A.E., Doak, D.F., Emmerson, M., Estes, J.A., *et al.* (2016). Characterizing species interactions to understand press perturbations: What is the community matrix? *Annu. Rev. Ecol. Evol. Syst.*, 47, 409–432.

Travis, C.C. & Post, W.M. (1979). Dynamics and comparative statics of mutualistic communities. *J. Theor. Biol.*, 78, 553–571.

5) I would also point out a technical issue of the direct usage of the S-map coefficients as IS. As is shown in Fig.4, it seems that the authors would intend to link the temporal variability of IS to the density changes of the recipient population. Here, please note that the temporal changes (and responses to disturbances) of IS include the changes in the population size of the recipient population. It implies the risk of statistical artifact of the relationship between IS changes and density changes. I hope this is my misunderstanding but the authors would want to make clearer this point.

As described above, we changed the use of model coefficients from population-level to individual-level coefficients. As such, we were able to avoid the possibility (risk) that the observed associations between the interaction effect and density (or the variability of both) are due to the proportional nature of the population-level interaction effect and density.

minor point

6) Line 447-448: "real time" --what does it mean? Would it be "every time"?

We corrected this phrase to “we obtained interaction effects not only at every time point” (LL 541-542).

Issues in storyline and arguments in Introduction and Discussion

1) With the usage of the S-map coefficients as the index of IS, I am not fully convinced by (i) the criticism on a 'static view' of interaction networks and classical mathematical ecology papers (e.g. May 1972) in Introduction (line 76-), and (ii) the argument on the possible mechanisms to realized temporal variability of IS (from Line 246), which addresses the flexible foraging. Including myself, I

am afraid that not a few scientists who apply EDM to estimate interaction strength oversell the temporal variability of S-map coefficients as an evidence of temporal variability of interaction strengths. Even with the 'static view' in the classical mathematical formulations, which use the temporally-constant interaction coefficients, the S-map coefficients changes with time under non-equilibrium state since the population size of the recipient species continuously changes with time. Thus I feel that the authors need to choose either of two alternative solutions; using the per-capita IS for the whole analyses with keeping the storyline or keeping using the current IS but dramatically softening the criticism (i) and the argument (ii) in the text.

As described above, we changed the use of model coefficients from population-level to individual-level coefficients and kept the overall storyline as much as possible. The choice of the individual-level interaction effect improved the justification of our study because the population-level interaction coefficient should change along with density changes under a nonequilibrium state, whereas the individual-level interaction coefficient should not change over time if a population model is formulated in a static manner. Therefore, the temporally varying individual-level interaction effect observed in our study is strong evidence that the effect of interactions is not static but rather highly variable, as noted in the Introduction. Furthermore, because the observed interaction variability occurred at the individual level, such variability may have resulted from behavioural changes such as flexible foraging.

2) Although I am sure that the results shown in Fig.4a-c have high novelty, I am not convinced at all by the authors'argument to link this "non-monotonic patterns" to "nonlinear response" of communities against disturbance and to the shift between buffering and amplifying effects by variabilities of IS. First, it is not acceptable to loosely use the term "nonlinear" or "nonlinearity" when just encountering nonlinear mathematical formulations. Please note that most of differential equation models and difference equation models of population dynamics use the nonlinear mathematical formulations to represent population growth and interspecific interactions but it does not imply the population dynamics governed by such "nonlinear" equations are nonlinear. Temporal fluctuations around the locally asymptotically stable equilibrium and those around the locally asymptotically stable limit cycles are evaluated as "linear" when applying univariate S-map; only when the attractors of the dynamics is torus or pseudo-strange attractors, the univariate S-map regards it as the nonlinear dynamics. It is confusing to use the term "nonlinear" in two distinct meanings in a manuscript that uses EDM analyses. I would say this is just a minor comment but more importantly, I do not think the non-monotonic patterns in Fig. 4a-c are related to the buffering nor amplification effects. When one wants to check the presence of buffering or amplifying effects of

a focal process (changes of IS in this study), it is necessary to compare the responses (density changes in this study) with the focal process and those without focal process as the baseline. However, in this study, since it would be difficult to predict or extrapolate the density responses without IS changes, I am afraid that the authors' argument is highly speculative and even misleading. It would be nicer if the authors could present more direct interpretation of this pattern in Discussion as well as in abstract.

We apologize for the ambiguous use of the term “nonlinear”. As the reviewer suggested, it is necessary to distinguish nonlinear mathematical formulations (e.g., polynomial functions) from nonlinear dynamics. In the previous version, we intended to explain that the shapes of the relationships were nonlinear in Fig. 4a-c (i.e., the former meaning). In the revision, to avoid potential confusion, we avoided using the term “nonlinear” as much as possible except for “nonlinear functional response” and “nonlinear time series analysis”, both of which are well-accepted technical terms.

Additionally, we understand that it is very difficult to test whether the interaction changes that actually occur mediate the density response to pesticides. As such, we focused on the effects of interaction density-dependence instead of those of interaction changes *per se*. Interaction density-dependence can be regarded as internal interaction properties or the propensity to change depending on density rather than just a realized interaction effect in response to environmental conditions. Therefore, interaction links whose effects show weaker density-dependence serve as controls appropriate for evaluating the effects of interaction variability on population instability. In the revision, the effects of interaction variability were evaluated by comparing the population instability against pesticide disturbances between corresponding interaction links whose interaction density-dependence was stronger or weaker. As a result, we found that negative density-dependent interactions tended to stabilize populations, whereas positive density-dependence tended to destabilize (Fig. 4a, e, i), which is consistent with a theoretical prediction proposed by previous studies (Kawatsu and Kondoh 2018; Kawatsu 2020). We speculate that the nonmonotonic, concave relationships between interaction change and population change observed in the previous version (Fig. 4a-c of the previous version) were generated by the mixture of the stabilizing and destabilizing effects of negative/positive interaction density-dependence.

Kawatsu, K. (2020). Ecology and evolution of density-dependence. In: *Diversity of Functional Traits and Interactions: Perspectives on Community Dynamics* (ed. Mougi, A.). Springer, Singapore, pp. 161–174.

Kawatsu, K. & Kondoh, M. (2018). Density-dependent interspecific interactions and the complexity – stability relationship. *Proc. R. Soc. B Biol. Sci.*, 285, 20180698.

Reviewer #2 (Remarks to the Author):

In this study, the authors explored whether the temporal flexibility of species interaction mitigates or amplifies the effect of anthropogenic disturbances on population dynamics. The detail can be summarized as follows.

Specifically, applying the EDM (empirical dynamic modelling) approach to the time series dataset of multitrophic mesocosms under four disturbance conditions (control, insecticide, herbicide and their combination), they quantified 1) the impact of disturbances on the density of the community members, 2) IS responses: changes in the magnitude of species interaction caused by disturbances and 3) IS intrinsic variability: temporal variability of species interaction under no disturbances. As a result, they found the nonlinear, concave relationship between IS responses and the impact of disturbances on the recipient density and the indirect path from IS intrinsic variability to the impact of disturbances via the IS responses. Based on these findings, they concluded that the effect of species interaction on the impact of anthropogenic disturbances depends on the interacting pairs but may be predictable by the information of the IS intrinsic variability.

Overall, the paper is well written: the text is easy to follow, and the methods/results are adequately described to be reproducible. I also agree with the ecological importance of the topic covered in this study. However, I have two concerns about the measure of interaction strengths used in this study.

Thank you for evaluating the potential of our manuscript and the comments about the measure of interaction effects. We switched the use of interaction effects from the population level to the individual level. This affects the analyses and the results substantially:

1. We removed the use of IS responses (changes in species interactions caused by disturbances) and used the degree of interaction density-dependence instead.
2. We stopped using the term 'IS intrinsic variability' because it consists of density-dependent and density-independent variability, and the latter may contain environment-driven variability. Thus, using the term 'intrinsic' was somewhat confusing.
3. The results of the concave relationship between IS responses and the impact of disturbances on population density were replaced. Instead, we evaluated the relationships between interaction density-dependence (IDD) and the impacts of disturbances on population density (population sensitivity to pesticides) and found that negative IDD tended to stabilize populations, whereas positive IDD tended to destabilize (Fig. 4a, e, i), which is consistent with a theoretical

prediction proposed by previous studies (Kawatsu and Kondoh 2018; Kawatsu 2020). We speculate that the nonmonotonic, concave relationships between interaction change and population change observed in the previous version were generated by the mixture of the stabilizing and destabilizing effects of negative/positive IDD.

Kawatsu, K. (2020). Ecology and evolution of density-dependence. In: *Diversity of Functional Traits and Interactions: Perspectives on Community Dynamics* (ed. Mougi, A.). Springer, Singapore, pp. 161–174.

Kawatsu, K. & Kondoh, M. (2018). Density-dependent interspecific interactions and the complexity – stability relationship. *Proc. R. Soc. B Biol. Sci.*, 285, 20180698.

(1) *First, the authors used the multivariate S-maps to quantify the strength of biological interactions; however, this method is known to be vulnerable to process noise, which includes the substantial variance in the estimated interaction strengths (Cenci et al. 2019). This point would be critical for this study: the indirect path from the variance of interaction strengths to the impact of disturbance (process noise) may be an artifact caused by the model's defect. Therefore, to remove this possibility, I suggest the authors reanalysed the data by use of the S-maps with regularization terms (a.k.a regularized S-map, Cenci et al. 2019).*

We reanalyzed our data by using the regularized S-map method. The results were not qualitatively different between the ordinally S-map and regularized one, and we reported the results generated by the regularized S-map in the revised version. We explained the regularized S-map method as follows: “Note that S-map is sensitive to data with stochastic noise, and we used a modified version of S-map, namely, regularized S-map, which incorporates a penalty term to avoid overfitting (Cenci et al. 2019). Regularized S-map has been shown to provide relatively robust estimations of interaction effects (Cenci et al. 2019).” (LL 536-539).

(2) *The second concern may be more critical. Specifically, S-map methods sequentially measured the Jacobian elements, which is mathematically identical to the partial derivative of the response variable with respect to the explanatory variable. Thus, if the raw values of population density are used as the model variables, the S-map coefficients corresponds to elements of the well-known ‘community matrix’ in ecological theory, which is the form as $X_i(t) \cdot \partial F_i / \partial X_j(t)$, in which $X_i(t)$, $X_j(t)$ and F_i stands for the density of recipient species at time t , that of donor species and the unknown function of per capita population growth of species i , respectively. As you see, the S-map coefficients*

obtained in this manner contain information on the density of the recipient species, making it difficult to use simple statistical modelling to analyze the relationship between the coefficients and the recipient's density. For example, it might be that IS responses did not mediate the impact of disturbances but simply that changes in the recipient's density were reflected in changes in the interaction strength. This defect cannot be resolved in the current S-map framework (Chang *et al.* 2021); however, the authors have to control information on the recipient's density, at least in their statistical analysis.

As suggested, we switched the use of model coefficients from population-level to individual-level coefficients by changing the response variable of S-map from the population growth rate to the *per capita* growth rate. This corresponds to the formal definition of *per capita* interaction effects formulated by Travis & Post (1979) and Novak *et al.* (2016): the first-order partial derivatives of a recipient's *per capita* growth rate on a donor density, $\partial(1/N_r \times dN_r/dt)/\partial N_d$, where N_r and N_d denote the density of the focal recipient and donor, respectively. As such, we were able to avoid the possibility (risk) that the observed associations between the interaction effect (or its variability) and density are due to the proportional nature of the population-level interaction effect and density.

We explained the detailed methods in the methods section as follows: “To determine the interaction effect at the *per capita* level, we took the *per capita* population growth rate as the response variable (Suzuki *et al.* 2017). The S-map coefficients estimated in this way theoretically correspond to the first-order partial derivatives of a recipient's *per capita* growth rate on a donor density: $\partial(1/N_r \times dN_r/dt)/\partial N_d$, where N_r and N_d denote the density of the focal recipient and donor, respectively (Travis & Post 1979; Novak *et al.* 2016). The *per capita* growth rate was approximated by calculating $\ln(N_{t+1}/N_t)$, where N_t and N_{t+1} are the raw density data of the focal species at time points t and $t+1$, respectively. Since there were several zero values in the raw density data, one was added to every density data point to calculate the *per capita* growth rate” (LL 526-536).

Novak, M., Yeakel, J.D., Noble, A.E., Doak, D.F., Emmerson, M., Estes, J.A., *et al.* (2016). Characterizing species interactions to understand press perturbations: What is the community matrix? *Annu. Rev. Ecol. Evol. Syst.*, 47, 409–432.

Travis, C.C. & Post, W.M. (1979). Dynamics and comparative statics of mutualistic communities. *J. Theor. Biol.*, 78, 553–571.

Reviewer #3 (Remarks to the Author):

The manuscript explores the effects of disturbance on interaction strength within mesocosm communities. They focus on pesticides as a disturbance, using both insecticide and herbicide, and show that variability in interaction strength can buffer the impacts of disturbance. However, this effect occurs only up to a certain disturbance threshold, beyond which variability in interaction strength amplifies the effects of disturbance. They combine experiments in mesocosm communities with empirical dynamic modeling to estimate interaction strength and understand the effect of disturbances in these communities.

1) One of the interesting aspects of this study is that in their experiments, they chose to use two different types of pesticides (insecticide and herbicide), which are common anthropogenic disturbances found in agricultural habitats (which is their interest). The authors find insecticide has a stronger effect on community members than herbicide. I do wonder whether this result is simply an effect of having more members in their mesocosm communities being susceptible to insecticide than herbicide. Meaning if most members are indeed insects or animals rather than plants or algae wouldn't the members from an animal kingdom be more directly affected by insecticide? This seems to be what they find, so it is not surprising but how much of their mesocosm is composed of insects?

We also suspect that the insecticide had a stronger impact than the herbicide simply because the communities consisted mainly of more insecticide-susceptible organisms, such as insects and arthropod animal plankton; therefore, the insecticide results having a stronger effect are a trivial finding. Nevertheless, because our analysis was completely revised, our argument that “IS responses to disturbances are more likely to amplify the disturbance impacts rather than buffer them when a disturbance is more severe, thereby impairing the resistance of the whole community” was also removed from the discussion.

2) One of the weaknesses I see with the manuscript is the authors' choice of monitoring community members as wide categories (functional groups). I understand they model effective interactions between community "guilds" but I do wonder how much they are missing in terms of interactions within each guild. Could some of the patterns that they observe in terms of variability in interaction strength be driven by dynamics happening within each category? How problematic is it to aggregate all herbivores (or other categories) of a given sub community and disregard all the interactions between members of the same category? Another point the authors briefly touch in their discussion is species turn over within each of these categories and how much of their results (including but not restricted to intrinsic variability) could be driven simply by species turnover. I am not sure I understand why fluctuations in interaction strength observed over short period of times rules out the

effects of species turn over.

Although previous studies describing interaction networks of freshwater communities have often aggregated individual species into single functional groups (e.g., Ives *et al.* 2003; Benincá *et al.* 2008; Matsuzaki *et al.* 2018), we recognize that such an approach is sometimes too simplistic. On the other hand, interactions among species within functional groups can be interpreted as being modelled by the effects of the density of the focal recipient functional group on its own *per capita* population growth rate in our S-map models. Thus, our analysis implicitly accounts for interactions between members of the same category. Nevertheless, it is true that some portion of the interaction variability could have arisen due to temporal species turnover, and we cannot refute this possibility.

We discussed this limitation in the *Methodological considerations* subsection as follows: “Second, estimating interaction effects by S-map models has been considered sensitive to the choice of variables (Ushio *et al.* 2018; Munch *et al.* 2022). This becomes more serious when the number of interacting variables is large because, in such a case, potential combinations of variables become increasingly larger, leading to unstable quantification of interaction effects (Chang *et al.* 2021). In our study, interaction effects were measured among the functional groups, not among the species. Although this simplification may be misleading because temporal species turnover within these groups could affect the observed interaction effects (Ives *et al.* 1999), it reduced the number of variables and hence the number of potential variable combinations. Thus, our estimation of interaction effects may be relatively robust. In addition, our observations were made over a two-week interval, and the interaction effects fluctuated over this relatively short time scale (Fig. S3b). If such a time scale was actually shorter than the rate of temporal species turnover within functional groups, the effects of temporal changes in species composition within functional groups on the observed interaction variability may have been relatively minor.” (LL 334-348).

Benincá, E., Huisman, J., Heerkloss, R., Jöhnk, K.D., Branco, P., Van Nes, E.H., *et al.* (2008). Chaos in a long-term experiment with a plankton community. *Nature*, 451, 822–825.

Ives, A.R., Dennis, B., Cottingham, K.L. & Carpenter, S.R. (2003). Estimating community stability and ecological interactions from time-series data. *Ecol. Monogr.*, 73, 301–330.

Matsuzaki, S.S., Suzuki, K., Kadoya, T., Nakagawa, M. & Takamura, N. (2018). Bottom-up linkages between primary production, zooplankton, and fish in a shallow, hypereutrophic lake. *Ecology*, 99, 2025–2036.

3) *One of your key results depends on the severity of the disturbance, meaning that as disturbance gets more severe, the effect of variability in interaction strength changes from buffer to*

amplification. From an empirical perspective, I wonder if the authors have a suggestion on how to estimate when a disturbance becomes severe? From their methodological point of view, if I understood correctly they gradually increased disturbance until it reached the threshold for which it changed from buffering to amplification? Would they still find the same result if instead of gradually increasing pesticide if they had different levels of pesticides to begin with? Meaning, are their results the same whether the disturbance is a press or pulse one?

We apologize for the unclear explanations in the discussion and methods section in the previous version. We did not manipulate the severity of disturbances *per se*, but manipulated the presence/absence of pesticides. In other words, the severity in this study was the result of experimental settings rather than the subjects of manipulation. In addition, we applied pesticides once a year, resulting in a pulse-like disturbance once a year. We explicitly emphasized this in the Methods section as follows: “Pesticide applications were performed at the beginning of the growing season of each experimental year (2017–2019), i.e., once a year” (LL 426-427). Nevertheless, because our analysis was completely revised, our argument about disturbance severity was also removed from the discussion.

4) The authors report changes in interaction strength over the years even in the control treatment which they refer to as an intrinsic characteristic of the community. I believe there are a few points related to this finding that require further exploration. It is not clear to me how exactly the authors effectively separate the effects of this intrinsic variability from the variability found in their experimental treatments. I understand they only have two replicates of each treatment and control but I do wonder if the replicates in the control changed in a similar way? Would this intrinsic variability be an effect of abiotic/environmental effects? If that is the case (environmental driven) would they expect similar changes to occur in the control with no pesticide as well as the other treatments equally? It would be interesting to discuss whether there were groups within the community for which intrinsic variability was greater than for others (for instance whether insects within their communities showed greater intrinsic interaction variability as they were possibly also the ones more heavily affected by the insecticide disturbance).

The interaction intrinsic variability (temporal standard deviation of interaction effect in the controls) was calculated only using the data of the controls; thus, it was independent of the disturbances caused by the pesticides. Therefore, even if the ‘interaction intrinsic variability’ of an insect is large, this variability is not due to insecticide impacts. However, the temporal standard deviation of the

interaction effect depends on both the properties intrinsic to the species involved and the environmental conditions due to the experimental settings, meaning that the ‘intrinsic variability’ was not separated from the variability in the experimental conditions. In the revised version, we partitioned the phenomenological temporal variability (i.e., what we referred to as intrinsic variability in the previous version) into density-dependent variability and density-independent variability, and we examined these effects separately (it becomes possible to perform such a partition by switching the use of interaction coefficients from the population level to the *per capita* level and describing the density-dependence of *per capita* interactions). We speculate that the former was generated by biological processes, whereas the latter may have included direct influences of environmental conditions, and we focused mainly on the former in the discussion in the revised version (“The observed IDD was mostly biologically interpretable, especially in cases of seemingly prey–predator interactions. For example,...”, LL 273-297). However, the actual mechanisms underlying both density-dependent and density-independent variability are unclear and need further investigation. In addition, we explored whether there were groups within the community for which interaction density-dependence was greater than for others by focusing on the biological functions of each group, but we failed to find consistent patterns and therefore did not include the discussion regarding this in the main text.

Additionally, we accept that the number of replicates was very low. We performed additional analyses to confirm whether the interaction effect showed similar dynamics between the replicates within the same treatments. For this purpose, we performed simple two-way ANOVA examining the relative contributions of the sources of variation in the S-map coefficients, i.e., treatment, census date, interaction and residuals. This analysis revealed that census date and treatment were the main sources of variation and that the effects of noise were minor, suggesting that the dynamics of the interaction effect were similar between the replicates. We explained this in the discussion and methods section as follows: “Finally, the number of replicates used in our experiment was low (i.e., 2). Previous ecotoxicological, mesocosm studies have also used a relatively small number of replicates (ranging from 2–5), probably due to resource and/or effort limitations (Hashimoto *et al.* 2019). Further studies are needed to confirm that the observed interaction dynamics are reproducible. Nevertheless, our additional analysis confirmed that the main source of variation (in a statistical sense) in the interaction effect was census date and treatment, not residuals (Fig. S5). Therefore, we believe that the dynamics of the interaction effect were relatively similar between the two replicates.” (LL 349-356). “To confirm whether the dynamics of the interaction effect were consistent among replicates, we performed two-way ANOVA for every interaction link, examining the effects of treatment, census date and their interaction on S-map coefficients. For simplicity, we did not account for any random effects or temporal autocorrelation” (LL 543-547).

Minor comments

5) Line 133: "disturbances become more severe" does this mean higher levels of pesticide? Is it incremental?

We apologize for the unclear explanations here. As mentioned above, this does not mean higher levels of pesticide but rather compares the effects between the insecticide, the herbicide and both. In addition, we applied the pesticides once a year. We explained this in the methods section as follows: "Pesticide applications were performed at the beginning of the growing season of each experimental year (2017–2019), i.e., once a year" (LL 426-427). In addition, the phrase "disturbances become more severe" was removed due to the substantial revision.

6) How consistent are the estimates of interaction strength across your replicates (not only control but pesticide treatments as well)?

As mentioned above, our additional analysis showed that the source of variation in the estimated interaction effects was mainly census date and treatment, suggesting that the estimation of interaction effects was consistent across replicates not only for the controls but also for the pesticide treatments (Fig. S5). We explained this in the discussion and methods section as follows: "Finally, the number of replicates used in our experiment was low (i.e., 2). Previous ecotoxicological, mesocosm studies have also used a relatively small number of replicates (ranging from 2–5), probably due to resource and/or effort limitations (Hashimoto *et al.* 2019). Further studies are needed to confirm that the observed interaction dynamics are reproducible. Nevertheless, our additional analysis confirmed that the main source of variation (in a statistical sense) in the interaction effect was census date and treatment, not residuals (Fig. S5). Therefore, we believe that the dynamics of the interaction effect were relatively similar between the two replicates." (LL 349-356).

7) Pesticide application (whether is gradually done or as a single application) needs further explanation. I understand this is a continuation of another study and that methods are fully explained there, but given that the severity of disturbance is a key point in their manuscript this needs to be clarified.

We apologize for the unclear explanations about the pesticide application method. As mentioned

above, we applied pesticides once a year. We explicitly emphasized this in the Methods section as follows: “Pesticide applications were performed at the beginning of the growing season of each experimental year (2017–2019), i.e., once a year” (LL 426-427).

***Updated figures**

Fig. 1. Schematic representation of three types of density-dependent (or density-independent) variability in the per capita interaction effect. This figure explains our specific predictions regarding the effects of interaction density (in)dependence on population stability.

Fig. 2. Pesticide impacts on the density of paddy community members. Panel a shows treatment effects on the density of each paddy community member, and no modifications were made. We added Panel b, which shows population sensitivity and is measured as the absolute value of the log response ratio of the mean raw density in the pesticide treatments (either I, H or I+H) relative to that in the controls for each of the three experimental years.

Fig. 3. Heterogeneous variability in per capita interaction effects within the experimental paddy community. Panels a-c show the interaction network, distribution of the mean interaction strength and interaction temporal variability, respectively, but updated by *per capita* interaction effects. We added Panel d, which shows density-dependence in *per capita* interaction effects in each interaction link.

Fig. 4. Stabilizing and destabilizing effects of different interaction properties on recipient population sensitivity to the three pesticide treatments. This figure shows how density-dependent and density-independent interaction variability mediate population sensitivity to pesticides (a proxy for population instability). Panels a, e, and i show that negative interaction density-dependence (IDD) tended to decrease population sensitivity, while positive IDD was more likely to increase population sensitivity. Panels b, f, and j show density-independent interaction variability increased population sensitivity.

REVIEWERS' COMMENTS and AUTHORS' RESPONSES

Reviewer #1 (Remarks to the Author):

The revision substantially considered my comments from the previous round of the reviewing process. I am happy to see that the authors' appropriately switched the index of interaction strength toward per capita based and developed a new idea of interaction density dependent (IDD). I am convinced by the new results showing the linkage between the sign of IDD, magnitude of IDD, and the population sensitivities against pesticides. However, I still have a concern on technical aspects on their study. The follows are the list of specific comments:

Thank you for reviewing and accepting most of our new idea and results. Below, we address your concern about our method of combining all the data and your minor comments.

Major comments: I am sorry that all the below are linked each other and somehow redundant each other.

1) I am not fully convinced by the method combining all data together to reconstruct a single common attractor while assuming (claiming) that interaction properties (such as mean interaction strength) were different among treatments.

More naturally, the authors could use 2 paddies x 3 years x 10 censuses 60 time points per each treatment to reconstruct treatment-specific attractor.

Or, did the authors let each paddy experience different treatments year by year?

The authors might need to technically justify this point.

Thank you for your critical suggestions regarding the method of combining all the data. Your suggestions (not only #1 but also #2 and #3) led us to more carefully consider the appropriateness of our method. First, we would like to describe the justification for this method.

Our method generally followed the guidelines suggested by Munch et al. (2022, "3.1 Short time series: Leveraging replicates" PP. 735-736), who mentioned that multiple time series showing similar dynamics can be combined to obtain better global predictions than would be obtainable using just a single short time series (Munch et al. 2022). This suggestion would support our approach of combining short time series from different treatments if we can consider that the dynamics of the

time series are sufficiently similar among different treatments. Furthermore, relevant approaches that combine similar but ecologically different (i.e., different species or experimental treatments) time series together for a single state-space reconstruction have been used by several previous studies (e.g., Hsieh et al. 2008; Virtanen et al. 2021) and introduced by the tutorial of the rEDM package (<https://ha0ye.github.io/rEDM/articles/rEDM.html#community-productivity-and-invasibility>).

In light of the above suggestion, we do not need to assume that a reconstructed attractor is exactly identical among different treatments (attached Fig. 1a). Rather, our method can be considered to make a prediction by using a *set* of reconstructed attractors that are assumed to share a common dynamical function (and thus share similar shapes) but may be characterized by different parameter values (and thus, their relative size and/or space they occupy would be different) (attached Fig. 1b). Assembling these different but similar attractors results in a single “thick” reconstructed attractor, the thickness of which corresponds to potentially different interaction properties and underlying parameter values of different treatments (attached Fig. 1b).

The strength of using such a set of attractors is that the data shortage in a single reconstructed attractor from each treatment (attached Fig. 1c) can be compensated for by other adjacent attractors, as long as the shapes of the attractors are sufficiently similar (attached Fig. 1d). Although we completely agree with the reviewer’s concern that it is more natural to use treatment-specific attractors, in our case, coping with information shortages would be a more critical issue. In fact, poor RMSE (approx. 2-6) values were shown by the S-map methods when treatment-specific attractors were used separately (attached Fig. 2). On the other hand, the RMSE was substantially decreased when all the data were combined (approx. < 0.1-1) (attached Fig. 2). This suggests that the shapes of attractors were sufficiently similar among different treatments; thus, we believe that the data combination successfully compensated for the data shortage of individual treatments.

To clarify the rationale of using our method more explicitly, we added a sentence and slightly modified the *Empirical Dynamic Modelling* subsection in the Methods section as follows: “EDM was originally designed to analyse single, relatively long time series (e.g., at least > 35–40 consecutive time points; Sugihara et al. 2012), yet several alternative approaches have been suggested for analysing shorter time series by combining multiple time series that are too short to be analysed individually (Hsieh et al. 2008, Clark et al. 2015, Munch et al. 2022). **Moreover, such a method can be applied to analyse multiple short time series belonging to ecologically different conditions (i.e., different species and/or experimental treatments) if the dynamics of short time series with different conditions are sufficiently similar (Hsieh et al. 2008, Virtanen et al. 2021).** Following these approaches, we assembled time series from the different experimental paddies and different years, which consisted of 10 time points × 8 paddies × 3 years = 240 time points in total, and analysed these assembled time series as a whole for each of our EDM analyses. Specifically, we allowed a single reconstructed attractor to consist of the whole data of every replicate, treatment and

year, while we constrained every single data point on the attractor to be generated only from the data of the same replicate in the same year. This approach implicitly assumes that the time series of different replicates, treatments and years share **dynamics that are sufficiently similar to analyse them together.**" (LL 466-481)

- Clark, A. T. et al. Spatial convergent cross mapping to detect causal relationships from short time series. *Ecology* 96, 1174–1181 (2015).
- Hsieh, C., Anderson, C. & Sugihara, G. Extending nonlinear analysis to short ecological time series. *Am. Nat.* 171, 71–80 (2008).
- Munch, S. B., Rogers, T. L. & Sugihara, G. Recent developments in empirical dynamic modelling. *Methods Ecol. Evol.* 14, 732–745 (2022).
- Virtanen, R., Clark, A. T., den Herder, M. & Roininen, H. Dynamic effects of insect herbivory and climate on tundra shrub growth: Roles of browsing and ramet age. *J. Ecol.* 109, 1250–1262 (2021).

2) For a technical justification, only when the forecasting ability from a single attractor and those from treatment-specific attractor are not different the authors' method would be fine. If the forecasting ability is higher in a single attractor, it implies that the all treatments are under the identical dynamical system. It is apparently nice but how can the authors claim that the interaction properties were different between treatments on a single common attractor? If the forecasting ability is lower, it implies that multiple attractors are mixed and interfered in the reconstruction and that the time series from different treatments belong to different attractor. Then, why not using the treatment-specific attractors?

Following your suggestion, we compared the forecasting ability between combined attractors (a set of treatment-specific attractors) and treatment-specific attractors alone by the RMSE values of the S-map models (attached Fig. 2). Using combined attractors substantially decreased the RMSE values consistently relative to those of treatment-specific attractors (i.e., combined attractors showed greater forecasting ability) (attached Fig. 2), implying that all the treatments had *similar* dynamics. Again, we would like to note that we do not need to assume that treatment-specific attractors are exactly the same among different treatments. As discussed in the response to your comment #1, the information shortage of one treatment-specific attractor can be compensated for by other adjacent, different but similar attractors (attached Fig. 1d). Moreover, since treatment-specific attractors are not necessarily identical among different treatments, implying that all the treatments are not necessarily under the identical dynamical system, the interaction properties should also differ among the treatments

(attached Fig. 1b). Consequently, we decided that we did not need to change our method.

3) It is difficult for me to imagine the situation where the time series from different treatments tend to stay longer in treatment-specific parts of the common attractor, resulting in the situation in which distinct interaction networks with different interaction magnitude were realized (Fig. S2). It is much more straightforward to reconstruct treatment-specific attractors and compared the interaction properties, in order to demonstrate such treatment-specific interaction properties.

As appropriately noted by the reviewer, distinct interaction networks in different treatments were realized if the time series from different treatments tended to stay longer in treatment-specific parts of the combined attractor (in other words, a *set* of treatment-specific attractors). This situation can easily occur when treatment-specific attractors are characterized by different parameter values of the common dynamical function. When the dynamical function underlying a time series is shared but its parameter values differ among different treatments, each time series tends to be attracted to a similar but different, treatment-specific attractor. In this way, the time series of different treatments can tend to stay longer in and/or around treatment-specific attractors, resulting in realized, distinct interaction networks (attached Fig. 1b).

In addition, as explained in the reply to your comment #1, the reconstructed treatment-specific attractors showed poor forecasting ability (attached Fig. 2), implying that the time series data of each individual treatment alone did not have sufficient information to reconstruct the dynamics. We consider that interaction effects estimated by such limited information might provide less reliable conclusions. Thus, as far as our situation is concerned, combining all the data together to leverage similar dynamic information would be the most appropriate method.

Minor comment:

Line 235: The statement is confusing. Why not using more direct statement, e.g. "greater magnitude of negative IDD resulted in smaller recipient sensitivity to pesticide disturbance"?

We rephrased this to "**We observed that a greater magnitude of negative IDD tended to result in lower recipient sensitivity to pesticide disturbance**". (LL 227-228)

Figure 3d: The color (red or blue) implies that there were no interaction pairs with density independent IDD?

The colour in this panel is assigned purely on the basis of the sign of the regression coefficients and is not affected by the statistical significance of the regressions. Thus, all the interaction pairs were assigned either of the two colours. We explained this in the caption of Fig. 3: **“For this colouration, we did not consider the statistical significance of the regressions”**. (LL 846-847)

Fig.4: I could not fully understand the data structure even if I carefully read the method part. Based on the results shown in Fig.3d, the sign of IDD (positive or negative) and the magnitude of the IDD (by the absolute value of the regression slope) were calculated in 23 pairs of the interactions. On the other hand, there apparently exist more than 23 data points in Fig.4, e.g. panels a, e, and I. Is this because with a single interaction density dependence value, there are three years' replicates?

We apologize for the insufficient explanations here. As you noted, this is because there are three years of replicates with a single interaction property value. We explained this in the caption of Fig. 4: **“The number of dots is three times greater than the number of interaction pairs (23) because there are three years of replicates with a single interaction property value”**. (LL 867-868)

Attached Fig. 1. Four situations of attractor reconstruction when analysing different treatments together. Descriptions are shown in each panel.

Attached Fig. 2. Forecasting skills of S-map models using combined and single reconstructed attractors. The RMSE values are shown. Cont: controls, Fipro: insecticide only, Pent: herbicide only, Joint: both insecticide and herbicide.

REVIEWERS' COMMENTS and AUTHORS' RESPONSES

Thank you for the detailed explanation of the suitability of your methodological approach. I would strongly encourage you to include the associated logic and attached figures 1 and 2 in your supplementary information so other readers can benefit from this understanding (with a cross-reference in the main text).

Thank you for accepting our explanation of the rationale of our data-combining method. Following your advice, we included a detailed explanation of the suitability of our methodology and the two figures from the rebuttal letter in the Supplementary materials as 'Supplementary Note 2' with a reference in the main text (L494 and L503).